# Latent-to-Data Cascaded Diffusion Models for Unconditional Time Series Generation

**Lifeng Shen**[1,2]**, Kai Syun Hou**[2]**, Weiyu Chen**[2]**, James T. Kwok**[2]
[1] Chongqing Key Laboratory of Computational Intelligence,
  Chongqing University of Posts and Telecommunications
[2] Department of Computer Science and Engineering,
  Hong Kong University of Science and Technology
`shenlf@cqupt.edu.cn, kshou@connect.ust.hk, wchenbx@cse.ust.hk,`
`jamesk@cse.ust.hk`

## Abstract

Synthetic time series generation (TSG) is crucial for applications such as privacy preservation, data augmentation, and anomaly detection. A key challenge in TSG lies in modeling the multi-modal distributions of time series, which requires simultaneously capturing diverse high-level representation distributions and preserving local temporal fidelity. Most existing diffusion models, however, are constrained by their single-space focus: latent-space models capture representation distributions but often compromise local fidelity, while data-space models preserve local details in the data space but struggle to learn high-level representations essential for multi-modal time series. To address these limitations, we propose L2D-Diff, a dual-space diffusion framework for synthetic time series generation. Specifically, L2D-Diff first compresses input sequences into a latent space to efficiently model the distribution of time series representations. The distribution then guides a data-space diffusion model to refine local data details, enabling faithful generation of time series distribution without relying on external conditions. Experiments on both single-modal and multi-modal datasets demonstrate the effectiveness of L2D-Diff in tackling unconditional TSG tasks. Ablation studies further highlight the necessity and impact of its dual-space design, showcasing its capability to achieve representation coherence and local fidelity.

## 1 Introduction

Time series data is critical in domains such as finance, healthcare, biotechnology, and climate science. However, restricted access to temporal datasets, especially in privacy-sensitive contexts, often limits the progress of machine learning models. Synthetic time series generation (TSG) has emerged as a promising solution, leveraging deep learning techniques to create realistic data that replicates complex temporal dependencies and multidimensional correlations (Zhou et al., 2023; Alaa et al., 2021; Ang et al., 2023; Yuan & Qiao, 2024). These synthetic datasets retain their utility for downstream tasks such as classification and forecasting (Esteban et al., 2017; Ang et al., 2023; Yuan & Qiao, 2024).

Generative adversarial networks (GANs) (Goodfellow et al., 2014) were the preferred approach for TSG (Esteban et al., 2017; Li et al., 2022; Mogren, 2016; Pei et al., 2021; Yoon et al., 2019). Despite their success, GANs face challenges such as adversarial training instability and mode collapse, limiting their effectiveness in generating diverse and robust time series. Recently, diffusion models (Yang et al., 2023), particularly the denoising diffusion probabilistic model (DDPM) (Ho et al., 2020), have gained prominence due to their superior perceptual quality and stable training dynamics. These advancements have led to significant progress in generative AI tasks (Yang et al., 2023), with diffusion models excelling in areas such as image editing (Huang et al., 2024), image generation (Cao et al., 2024), and video generation (Xing et al., 2024).

While diffusion models have achieved remarkable success in images and videos, their application to time series presents unique challenges. Unlike visual data, time series generation requires the simultaneous modeling of multi-modal latent structures and the preservation of local temporal fidelity.

This involves capturing intricate temporal relationships and managing complex interdependencies across variables, both of which are essential for accurately modeling real-world time series patterns. Addressing these challenges is crucial for extending the capabilities of diffusion models to TSG.

Recent works on time series diffusion primarily focus on conditional generation tasks such as forecasting (Rasul et al., 2021; Shen & Kwok, 2023; Kollovieh et al., 2024) and imputation (Tashiro et al., 2021; Alcaraz & Strodthoff, 2022). For instance, TimeGrad (Rasul et al., 2021) employs recurrent neural networks to summarize history as conditions for denoising future values. Similarly, TimeDiff (Shen & Kwok, 2023) introduces autoregressive initialization and future mixup to enable efficient non-autoregressive prediction. CSDI (Tashiro et al., 2021) adopts self-supervised masking techniques, while Alcaraz & Strodthoff (2022) enhance CSDI by replacing transformers with structural state space models (Gu et al., 2021), improving long-range temporal modeling. These studies primarily focus on leveraging conditional information, designing robust conditioning networks, and constructing effective denoising architectures to address specific supervised tasks. In contrast, synthetic time series generation focuses on unconditionally producing high-quality time series (modeling the data distributions) that replicate the statistical properties of the original dataset (Ang et al., 2023).

Recent approaches to unconditional generation can be broadly divided into two categories. *(i) Data-space diffusion models* (Park et al., 2024; Yuan & Qiao, 2024; Crabbé et al., 2024; Naiman et al., 2024a; Sikder et al., 2025), which directly model the raw time series distribution; and *(ii) latent-space diffusion models* (Qian et al., 2024; Feng et al., 2024), which operate on compressed representations obtained through predefined transformations (e.g., Fourier transform) or learned nonlinear encoders. While latent-space diffusion models excel at capturing high-level semantic structures through compressed representations, they often struggle to preserve subtle temporal dynamics. The process of dimensionality reduction can result in the loss of fine-grained details, thereby reducing the diversity and fidelity of the generated outputs. On the other hand, data-space diffusion models perform iterative denoising directly on the raw time series, effectively capturing localized temporal patterns with high precision. However, their focus on local details makes it difficult to comprehensively model representation distributions.

To address these challenges, we transition from unconditional diffusion in the data space to latent-to-data conditional diffusion, which balances representation distributions with local temporal data distributions. Specifically, we propose L2D-Diff, a latent-to-data diffusion framework that integrates the strengths of latent-space modeling and data-space refinement to overcome the limitations of unconditional generation. L2D-Diff operates in two complementary stages: *(i) Latent-space coarse generation:* A latent diffusion model captures representation distributions by representation learning techniques. *(ii) Data-space refinement:* A subsequent denoising process integrates the global latent codes into the data space, enabling fine-grained temporal precision and ensuring consistency with the original data distribution. This two-stage approach ensures both global consistency and local precision, enabling realistic, semantically rich, and high-fidelity time series generation. To the best of our knowledge, we are the first to study the latent-to-data cascaded diffusion model for synthetic time series generation.

|  | data | latent | cascaded |
|---|---|---|---|
| TimeGrad | ✓ | ✗ | ✗ |
| CSDI | ✓ | ✗ | ✗ |
| TimeDiff | ✓ | ✗ | ✗ |
| TSDE | ✓ | ✗ | ✗ |
| TimeLDM | ✗ | ✓ | ✗ |
| LDT | ✗ | ✓ | ✗ |
| DiffusionTS | ✓ | ✗ | ✗ |
| L2D-Diff (proposed) | ✓ | ✓ | ✓ |

Table 1: Comparing related diffusion methods. "data" refers to directly modeling the time series distribution in the data space. "latent" indicates learning the distribution of representations in a latent space. "cascaded" denotes using multiple diffusion models for generation.

Some initial attempts have been proposed in the contexts of image generation and graph modeling. For example, in the representation-conditioned generation (RCG) framework (Li et al., 2024), a pre-trained image encoder is used to first obtain image representation distributions, which then condition the image distributions. This is further extended to the generation of graphs in (Wang et al., 2024). Another model EDDPM (Liu et al., 2019) uses parameterized encoding-decoding in a unified space to generalize the Gaussian noising-denoising in standard data-space diffusion. However, the development of hybrid models for time series generation is still under-explored. In (Ge et al., 2025), a text-to-series diffusion model (T2S) is developed that leverages textual features to assist in time series generation. In contrast to T2S, we do not utilize additional textual descriptions. Our

focus is on establishing an effective representation-to-data cascaded model for unconditional time series generation. The advantage of our approach lies in its independence from the text processing mechanisms of large language models, making it simpler and more efficient.

Note that some cascaded time series diffusion models exist, including (Fan et al., 2024; Shen et al., 2024). However, these models are not originally designed for unconditional generation. In contrast, the proposed L2D-Diff formulates a cascade between a latent-space diffusion model and a data-space diffusion model, and is, to the best of our knowledge, the first framework to explicitly bridge latent representation diffusion and data-space diffusion for unconditional time series generation. Table 1 provides a comparison between the proposed method and related works.

## 2 RELATED WORKS

This section reviews diffusion models for unconditional time series generation. Classical generative models, such as GANs, VAEs, and flow-based approaches, are covered in Appendix A.

Data-space diffusion models directly model the raw time series distribution. Park et al. (2024) employ diffusion bridges to map prior distributions to time series, enabling flexible and accurate synthesis. Diffusion-TS (Yuan & Qiao, 2024) integrates seasonal-trend decomposition with diffusion models and introduces a Fourier-based objective to better capture periodic patterns. Similarly, FourierDiffusion (Crabbé et al., 2024) operates in the frequency domain, replacing the traditional Brownian motion with mirrored Brownian motion to enhance its ability to model periodic behaviors. Naiman et al. (2024a) transform time series into images and apply vision-based diffusion models to synthesize data. Sikder et al. (2025) recently develop TransFusion for long sequence generation.

Latent-space diffusion models operate on compressed representations obtained through predefined transformations (e.g., Fourier transform) or learned nonlinear encoders. Representative methods such as TimeLDM (Qian et al., 2024) and latent diffusion transformer (LDT) (Feng et al., 2024) achieve computational efficiency by working in a lower-dimensional latent space. This compression helps preserve structural representations within the data distributions. However, the reliance on encoder-decoder architectures introduces an information bottleneck, which risks discarding fine-grained temporal details and limits the fidelity of the generated outputs.

## 3 PRELIMINARIES

**Problem definition.** Let $\mathcal{T} = \{\mathbf{x}^{(1)}, \ldots, \mathbf{x}^{(N)}\}$ be a dataset with $N$ multivariate time series samples. Each $\mathbf{x}^{(i)} = (\mathbf{x}_1^{(i)}, \ldots, \mathbf{x}_L^{(i)})$, with $\mathbf{x}_t^{(i)} \in \mathbb{R}^D$, can be represented as a $D$-by-$L$ matrix, where $D$ is the number of variables and $L$ is the time series length. The goal of synthetic time series generation (TSG) is to *create a synthetic dataset* $\mathcal{T}^{gen} = \{\tilde{\mathbf{x}}^{(1)}, \ldots, \tilde{\mathbf{x}}^{(N')}\}$ such that its distribution $q(\mathcal{T}^{gen})$ is similar to the true distribution $p(\mathcal{T})$, exhibiting consistent statistical properties and temporal dynamics. This is an unconditional generation task. Importantly, we require each synthetic time series $\tilde{\mathbf{x}}^{(i)}$ to be also of length $L$ and contain $D$ variables, ensuring compatibility with the original dataset structure.

### 3.1 DENOISING DIFFUSION PROBABILISTIC MODELS

Denoising diffusion probabilistic model (DDPM) (Ho et al., 2020) is a latent variable model with forward diffusion and backward denoising processes.

**Forward diffusion.** A time series input[1] $\mathbf{x}^0$ is gradually corrupted to a Gaussian noise vector. At the $k$th step, $\mathbf{x}^k$ is generated by corrupting the previous iterate $\mathbf{x}^{k-1}$ (scaled by $\sqrt{1-\beta_k}$) with zero-mean Gaussian noise (with variance $\beta_k \in [0,1]$):

$$q(\mathbf{x}^k|\mathbf{x}^{k-1}) = \mathcal{N}(\mathbf{x}^k; \sqrt{1-\beta_k}\mathbf{x}^{k-1}, \beta_k\mathbf{I}), \quad k = 1, \ldots, K.$$

It can be shown that $q(\mathbf{x}^k|\mathbf{x}^0) = \mathcal{N}(\mathbf{x}^k; \sqrt{\bar{\alpha}_k}\mathbf{x}^0, (1-\bar{\alpha}_k)\mathbf{I})$, where $\bar{\alpha}_k = \Pi_{s=1}^k \alpha_s$, and $\alpha_k = 1-\beta_k$. Thus, $\mathbf{x}^k$ can be simply obtained as

$$\mathbf{x}^k = \sqrt{\bar{\alpha}_k}\mathbf{x}^0 + \sqrt{1-\bar{\alpha}_k}\epsilon, \tag{1}$$

---

[1] Here, superscript 0 means the original input without diffusion.

where $\epsilon$ is a noise from $\mathcal{N}(\mathbf{0}, \mathbf{I})$. This equation also allows $\mathbf{x}^0$ to be easily recovered from $\mathbf{x}^k$.

**Reverse denoising.** At the $k$th denoising step, $\mathbf{x}^{k-1}$ is generated from $\mathbf{x}^k$ by sampling from the normal distribution:

$$p_\theta(\mathbf{x}^{k-1}|\mathbf{x}^k) = \mathcal{N}(\mathbf{x}^{k-1}; \mu_\theta(\mathbf{x}^k, k), \Sigma_\theta(\mathbf{x}^k, k)). \tag{2}$$

Here, the variance $\Sigma_\theta(\mathbf{x}^k, k)$ is usually fixed as $\sigma_k^2 \mathbf{I}$, while the mean $\mu_\theta(\mathbf{x}^k, k)$ is defined by a neural network (parameterized by $\theta$). This is usually formulated as a noise estimation or data prediction problem (Benny & Wolf, 2022). For noise estimation, a network $\epsilon_\theta$ predicts the noise of the diffused input $\mathbf{x}^k$, and then obtains $\mu_\theta(\mathbf{x}^k, k) = \frac{1}{\sqrt{\alpha_k}}\mathbf{x}^k - \frac{\beta_k}{\sqrt{\alpha_k}\sqrt{1-\bar{\alpha}_k}}\epsilon_\theta(\mathbf{x}^k, k)$. Parameter $\theta$ is learned by minimizing the noise estimation loss $\mathcal{L}_\epsilon = \mathbb{E}_{k,\mathbf{x}^0,\epsilon}\left[\|\epsilon - \epsilon_\theta(\mathbf{x}^k, k)\|^2\right]$.

Alternatively, the data prediction strategy uses a denoising network $\mathbf{x}_\theta(\cdot)$ to obtain an estimate $\mathbf{x}_\theta(\mathbf{x}^k, k)$ of the clean data $\mathbf{x}^0$ given $\mathbf{x}^k$, and then set

$$\mu_\theta(\mathbf{x}^k, k) = \frac{\sqrt{\alpha_k}(1 - \bar{\alpha}_{k-1})}{1 - \bar{\alpha}_k}\mathbf{x}^k + \frac{\beta_k\sqrt{\alpha_k}}{1 - \bar{\alpha}_k}\mathbf{x}_\theta(\mathbf{x}^k, k). \tag{3}$$

Then, $\theta$ is learned by minimizing the following loss

$$\mathcal{L}_\mathbf{x} = \mathbb{E}_{\mathbf{x}^0,\epsilon,k}\|\mathbf{x}^0 - \mathbf{x}_\theta(\mathbf{x}^k, k)\|^2. \tag{4}$$

When a condition $\mathbf{c}$ is accessible, the following distribution is considered (Rasul et al., 2021; Tashiro et al., 2021; Shen & Kwok, 2023)

$$p_\theta(\mathbf{x}^{0:K}|\mathbf{c}) = p(\mathbf{x}^K)\prod_{k=1}^{K}p_\theta(\mathbf{x}^{k-1}|\mathbf{x}^k, \mathcal{F}(\mathbf{c})), \tag{5}$$

where $\mathbf{x}^K \sim \mathcal{N}(\mathbf{0}, \mathbf{I})$, and $\mathcal{F}$ is a conditioning network that takes the condition $\mathbf{c}$ as input. Correspondingly, the denoising process at step $k$ is

$$p_\theta(\mathbf{x}^{k-1}|\mathbf{x}^k, \mathbf{c}) = \mathcal{N}(\mathbf{x}^{k-1}; \mu_\theta(\mathbf{x}^k, k|\mathcal{F}(\mathbf{c})), \sigma_k^2 \mathbf{I}), \tag{6}$$

where $k = K, K-1, \ldots, 1$.

**Data sampling.** During inference, let the generated sample corresponding to $\mathbf{x}^k$ be $\hat{\mathbf{x}}^k$. We first initialize $\hat{\mathbf{x}}^K$ as a noise vector from $\mathcal{N}(\mathbf{0}, \mathbf{I})$. By repeatedly running the denoising step in Equation (6) till $k = 1$, the final generation is $\hat{\mathbf{x}}^0$.

## 3.2 LATENT-SPACE DIFFUSION MODELS

Latent-space diffusion models (LDMs) (Rombach et al., 2022) consist of two main components: (i) pretraining process and (ii) latent diffusion. The pretraining process is commonly performed based on optimizing a representation learning task, such as masked modeling or contrastive learning. It involves an encoder, which maps time series $\mathbf{x} \in \mathbb{R}^{D \times L}$ to a lower-dimensional (fixed-length) latent space $\mathbf{r} \in \mathbb{R}^d$, and a decoder, which generates $\mathbf{x}$ from $\mathbf{r}$. Subsequently, a diffusion model is applied on the latent code $\mathbf{r}$. The forward diffusion process in latent space is:

$$\mathbf{r}^k = \sqrt{\bar{\alpha}_k}\mathbf{r}^0 + \sqrt{1 - \bar{\alpha}_k}\epsilon. \tag{7}$$

The reverse process is performed by recursively sampling $\mathbf{r}^{k-1}$ from the following normal distribution:

$$p_\phi(\mathbf{r}^{k-1}|\mathbf{r}^k) = \mathcal{N}(\mathbf{r}^{k-1}; \mu_\phi(\mathbf{r}^k, k), \Sigma_\phi(\mathbf{r}^k, k)). \tag{8}$$

The training objective minimizes the following loss:

$$\mathcal{L}_\mathbf{r} = \mathbb{E}_{\mathbf{r}^0,\epsilon,k}\|\mathbf{r}^0 - \mathbf{r}_\phi(\mathbf{r}^k, k)\|^2. \tag{9}$$

Here, $\mathbf{r}_\phi$ is a denoising network in the latent space.

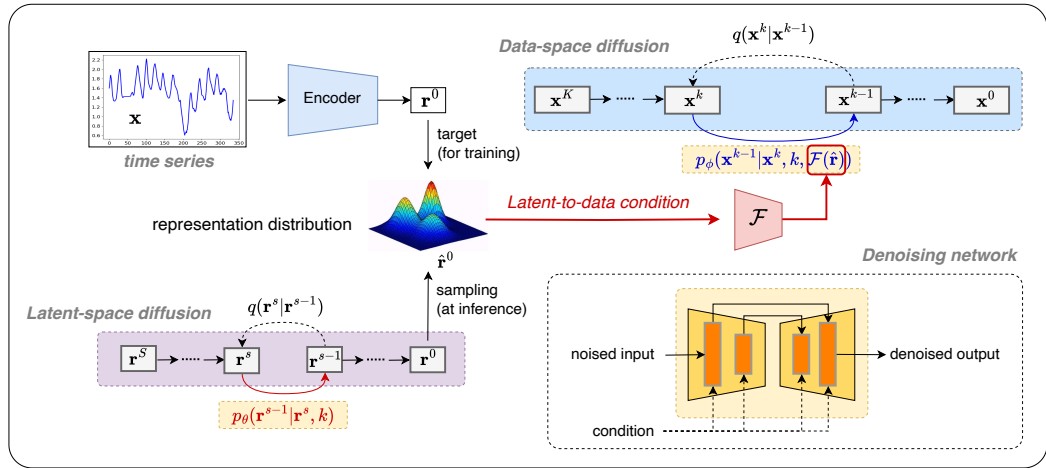

Figure 1: Framework of the proposed L2D-Diff for time series generation.

## 4 METHODOLOGY

**Overview.** The proposed cascaded diffusion model, L2D-Diff, is illustrated in Figure 1. It integrates two collaborative diffusion/denoising branches: one in the latent space and the other in the data space. The latent-space branch models the distribution of high-level representations in the time series, offering a compressed yet structured understanding of temporal patterns. To construct the latent space, an encoder-decoder pair is pretrained using masked modeling-based representation learning optimization, ensuring that the latent representations are meaningful and informative. The data-space branch then models the probability density function of the time series data guided by the representation distributions, capturing fine-grained temporal details. To bridge these two branches, a latent-to-data conditioning mechanism is introduced. This module enables latent representations to guide the denoising process in the data space, ensuring seamless coordination between the representation distribution and data distribution.

This design enables L2D-Diff to effectively capture high-level temporal patterns in the latent space while achieving data detail generation in the data space, guided by the latent variables. In Appendix B, we provide a theoretical understanding of L2D-Diff through the lens of the Information Bottleneck (IB) principle (Tishby et al., 1999). Intuitively, our latent-to-data cascaded structure presents a *divide-and-conquer* strategy. With a rich latent space, L2D-Diff utilizes latent diffusion to capture high-level semantics of the data while allowing the data diffusion process to more easily focus on modeling local details and residual uncertainties, resulting in higher-quality generation.

Although there have been attempts to study representation-conditioned generation on data modalities such as image and audio, time series generation presents unique challenges not fully addressed by existing cascaded diffusion frameworks, particularly concerning temporal consistency and multi-channel correlations. Time series data has inherent temporal dependencies, meaning that the sequence of data points affects future values. Ensuring temporal consistency is crucial, as high-quality generated series must reflect realistic trends and relationships. Moreover, many time series comprise multiple channels that may interact in complex ways, necessitating the capture of these correlations to produce coherent and high-quality outputs. By addressing these challenges, our work aims to provide a robust framework for unconditional time series generation, ultimately raising the quality standards for generated time series with multiple modes that leverage their full potential.

### 4.1 DUAL-BRANCH DIFFUSION DESIGN

**Latent Space Construction.** Given an input time series $\mathbf{x} \in \mathbb{R}^{D \times L}$, where $D$ is the number of channels and $L$ the sequence length, we perform a pretraining task based on masked modeling to derive a compact, high-level representation $\mathbf{r} \in \mathbb{R}^d$, with $d \ll L \times D$.

In this process, a random subset of the input tokens are masked according to a binary mask $\mathbf{m} \in \{0,1\}^{D \times L}$, where $m_{i,j} = 1$ indicates that the token $(i,j)$ is masked. The masked input $\mathbf{x}_{\text{masked}}$ is obtained by replacing the masked positions with special mask tokens. The encoder $\mathbf{E}$ processes the corrupted input $\mathbf{x}_{\text{masked}}$, generating a latent representation $\mathbf{r}_{\text{masked}} = \mathbf{E}(\mathbf{x}_{\text{masked}})$. The decoder $\mathbf{D}$ reconstructs the original input $\mathbf{x}$ from the latent representation $\mathbf{r}_{\text{masked}}$. The optimization objective is designed to minimize the reconstruction error at the masked positions only: $\mathcal{L}_{\text{pretraining}} = \|\mathbf{m} \odot (\mathbf{x} - \mathbf{D}(\mathbf{E}(\mathbf{x}_{\text{masked}})))\|_2^2$, where $\odot$ denotes element-wise multiplication, ensuring that only the masked positions contribute to the loss.

**Latent-Space Diffusion.** After pretraining, input $\mathbf{x}$ is encoded into the representation $\mathbf{r} = \mathbf{E}(\mathbf{x})$. Intuitively, $\mathbf{r}$ encapsulates the high-level temporal characteristics of $\mathbf{x}$. We then introduce a latent-space diffusion model, denoted $\mathbf{r}_\phi$ (where $\phi$ denotes its parameters), to model the distribution of $\mathbf{r}$ over $S$ diffusion steps. The diffused representation $\mathbf{r}^s$ is obtained from $\mathbf{r}^0 (= \mathbf{r})$ following (7):

$$\mathbf{r}^s = \sqrt{\bar{\alpha}_s}\mathbf{r}^0 + \sqrt{1 - \bar{\alpha}_s}\epsilon, \tag{10}$$

where $\epsilon$ is the Gaussian noise, $\bar{\alpha}_s$ governs the noise level at step $s$ ($1 \leq s \leq S$), and $S$ is the total number of latent diffusion steps.

To train $\mathbf{r}_\phi$, we minimize the denoising loss in (9), which encourages $\mathbf{r}_\phi$ to recover the original representation $\mathbf{r}^0$ from its noisy counterpart $\mathbf{r}^s$:

$$\mathcal{L}_{latent} = \mathbb{E}_{\mathbf{r}^0,\epsilon,s}\|\mathbf{r}^0 - \mathbf{r}_\phi(\mathbf{r}^s, s)\|^2. \tag{11}$$

**Data-Space Diffusion.** The data-space diffusion model regenerates the full-resolution series $\mathbf{x} \in \mathbb{R}^{D \times L}$, guided by the representation encoded in the latent space. This latent-to-data diffusion mechanism allows each position in the data-space series $\mathbf{x}_t$ to attend to the latent code $\mathbf{r}$, effectively injecting structural priors into local refinements.

Following (4), the diffusion model is optimized by minimizing the denoising loss

$$\mathcal{L}_{data} = \mathbb{E}_{\mathbf{x}^0,\epsilon,k}\|\mathbf{x}^0 - \mathbf{x}_\theta(\mathbf{x}^k, k, \mathcal{F}(\mathbf{c}))\|^2, \tag{12}$$

where $\mathbf{x}_\theta$ is the denoising network, $\mathbf{c}$ is the condition, and $\mathcal{F}$ is the conditioning network. In practice, $\mathcal{F}$ is implemented as a convolutional neural network (with 5 layers by default). At each denoising step $k$, $\mathbf{x}_\theta$ takes three inputs: noisy input $\mathbf{x}^k \in \mathbb{R}^{D \times L}$, timestep $k$ and the conditioning network's output $\mathcal{F}(\mathbf{c})$, to produce a data estimate $\mathbf{x}_\theta(\mathbf{x}^k, k, \mathcal{F}(\mathbf{c}))$.

## 4.2 LATENT-TO-DATA CONDITIONING

**Conditioning Network.** In L2D-Diff, unconditional time series generation is reformulated as a conditional generation problem, where the latent representation sampled from the learned latent distribution, denoted $\hat{\mathbf{r}}$, serves as the condition for the data-space diffusion model. In this way, the global structure encoded in the latent space is explicitly injected into the data-space denoising process, enabling the diffusion model to generate samples consistent with the learned representation distribution.

During training, the conditioning network $\mathcal{F}$ takes the latent code $\mathbf{r} \in \mathbb{R}^d$ as input, i.e., $\mathbf{c} = \mathbf{r}$, and transforms it into a conditioning signal compatible with the data-space diffusion model. Intuitively, $\mathcal{F}$ learns to project latent representations into a structured guidance signal that modulates the denoising trajectory at each diffusion step.

This latent-to-data conditioning mechanism ensures that the representation distribution captured in the latent space effectively guides the local refinements in the data space, leading to high-quality time series generation that aligns with both representation distribution and data representation.

**Denoising Network.** The denoising networks $\mathbf{r}_\phi$ in Equation (11) (reps. $\mathbf{x}_\theta$ in Equation (12)) is trained to learn to denoise the representation $\mathbf{r}^s$ (resp. the diffused data $\mathbf{x}^k$) into $\mathbf{r}^{s-1}$ (resp. $\mathbf{x}^{k-1}$). The key distinction between the two denoising networks lies in their conditioning mechanisms: $\mathbf{x}_\theta$ incorporates a latent-space-derived conditioning signal $\mathbf{r}$ to guide the refinement process.

### 4.3 Synthetic Time Series Generation

On inference, we start from $\hat{\mathbf{r}}^S \sim \mathcal{N}(\mathbf{0}, \mathbf{I})$ in the latent space. Based on the data prediction strategy in (3),

$$\hat{\mathbf{r}}^{s-1} = \frac{\sqrt{\alpha_s}(1 - \bar{\alpha}_{s-1})}{1 - \bar{\alpha}_s} \mathbf{r}^s + \frac{\sqrt{\bar{\alpha}_{s-1}}(1 - \alpha_s)}{1 - \bar{\alpha}_s} \mathbf{r}_\phi(\mathbf{r}^s, s) + \sigma_s \epsilon, \tag{13}$$

where $\epsilon \sim \mathcal{N}(\mathbf{0}, \mathbf{I})$ when $s > 1$, and $\epsilon = 0$ otherwise. Till $s = 1$, we obtain the sampled representation $\hat{\mathbf{r}}^0$. Then, we have $\mathbf{c} = \hat{\mathbf{r}}^0$ to guide the data denoising process. Specifically, we start from $\hat{\mathbf{x}}^K \sim \mathcal{N}(\mathbf{0}, \mathbf{I})$ in the data space, and we have the reverse denoising step equation

$$\hat{\mathbf{x}}^{k-1} = \frac{\sqrt{\alpha_k}(1 - \bar{\alpha}_{k-1})}{1 - \bar{\alpha}_k} \mathbf{x}^k + \frac{\sqrt{\bar{\alpha}_{k-1}}(1 - \alpha_k)}{1 - \bar{\alpha}_k} \mathbf{x}_\theta(\mathbf{x}^k, k, \mathcal{F}(\mathbf{c})) + \sigma_k \epsilon. \tag{14}$$

Till $k = 1$, we obtain the sampled time series data $\hat{\mathbf{x}}^0$. The pseudocodes for the training and sampling procedures are in Algorithms 1 and 2 of Appendix C, respectively.

## 5 Experiments

In this section, we perform experiments on a number of datasets to demonstrate the performance of the proposed L2D-Diff.

### 5.1 Setup

**Datasets.** Previous works, such as Diffusion-TS (Yuan & Qiao, 2024), mainly use time series datasets with relatively limited distributional diversity (*Stock*, *Energy*, *ETTh*, *Riverflow*). These datasets are typically constructed by using a sliding window over the whole time series. While temporal variations exist, these patterns are drawn from the same underlying data distribution, making them less suitable for evaluating performance across distinct labeled categories.

To better evaluate generative performance, we include seven more datasets with explicit class annotations[2] from the time series classification archive[3]. Hence, they have inter-class variability and the underlying data distributions exhibit multiple modes.

| dataset | #training | #testing | $D$ | $L$ | $C$ |
|---|---|---|---|---|---|
| *Stock* | 2,928 | 733 | 6 | 24 | - |
| *Energy* | 15,768 | 3,943 | 28 | 24 | - |
| *ETTh* | 13,801 | 3,451 | 7 | 168 | - |
| *Riverflow* | 18,858 | 4,715 | 1 | 168 | - |
| *Two Patterns* | 1,000 | 4,000 | 1 | 128 | 4 |
| *ECG5000* | 500 | 4,500 | 1 | 140 | 5 |
| *Medical Images* | 381 | 760 | 1 | 99 | 10 |
| *Arabic Digits* | 6,600 | 2,200 | 13 | 93 | 10 |
| *Atrial Fibrillation* | 4,832 | 185 | 2 | 45 | 3 |
| *Japanese Vowels* | 270 | 370 | 12 | 29 | 9 |
| *Character Trajectories* | 300 | 2,558 | 3 | 205 | 20 |

Table 2: Summary of dataset statistics ($D$: dimension; $L$: time series length; $C$: number of classes).

Table 2 shows statistics of the datasets. To ensure a fair evaluation of generalization in the unconditional generation setting, the data are uniformly split into training and test sets. The test samples are not used during training, preventing data leakage and enabling evaluation on unseen data.

**Baselines.** We include baselines from various categories: (i) Diffusion models operating in the time domain: Diffusion-TS (Yuan & Qiao, 2024) and TSDE (Senane et al., 2024); (ii) Latent diffusion models: TimeLDM (Park et al., 2024) and EDDPM (Liu et al., 2019); (iii) Diffusion models operating in the Fourier domain: Fourier Diffusion (Crabbé et al., 2024) and ImagenTime (Naiman et al., 2024a). (iv) Flow-based generative models: FourierFlow (Alaa et al., 2021) and its time-domain adaptation, referred to as TimeFlow, which applies the FourierFlow framework directly in the time domain for comparison. (v) Generative adversarial networks (GANs): We include two popular baselines in (Ang et al., 2023): TimeGAN (Yoon et al., 2019) and GTGAN (Jeon et al., 2022); (vi) Variational autoencoder (VAE) models, including KoVAE (Naiman et al., 2024b) TimeVQVAE (Lee et al., 2023) LS4 (Zhou et al., 2023) and the original VAE (Kingma & Welling, 2014).

---

[2]Though these datasets contain label information, such information is not used during the generation process.
[3]https://www.timeseriesclassification.com/

|  | Stock | Energy | ETTh | Riverflow | Two Patterns | ECG5000 | Medical Images | Arabic Digits | Atrial Fibrillation | Japanese Vowels | Character Trajectories |
|---|---|---|---|---|---|---|---|---|---|---|---|
| **L2D-Diff** | 0.31 | 0.53 | **0.45** | 0.32 | **0.21** | **0.11** | **0.08** | 1.29 | 1.15 | **0.46** | **0.28** |
| Diffusion-TS | 0.49 | 0.82 | 4.75 | 1.24 | 1.69 | 1.95 | 3.10 | 1.66 | 2.39 | 1.93 | 3.57 |
| TSDE | 3.90 | 4.13 | - | **0.26** | 2.83 | 1723.0 | 0.85 | 2.60 | - | 3.97 | 4.70 |
| TimeLDM | 6.17 | 3.51 | 9.52 | 1.01 | 1.40 | 0.76 | 0.88 | 5.99 | 5.48 | 0.99 | 2.00 |
| EDDPM | 2.31 | 2.89 | 10.76 | 28.29 | 6.72 | 1.11 | 1.01 | 5.40 | 4.63 | 1.40 | 3.56 |
| FourierDiffusion | **0.21** | 0.48 | 3.38 | 3.54 | 1.16 | 0.32 | 0.41 | **1.26** | **1.14** | 0.49 | 3.58 |
| ImagenTime | 4.23 | 2.22 | 7.72 | 0.50 | 4.82 | 6.38 | 2.99 | 2.98 | 1.66 | 1.08 | 12.02 |
| FourierFlow | 1.15 | **0.38** | 3.17 | 1.843 | 1.21 | 0.98 | 1.52 | 2.84 | 2.37 | 0.74 | 5.07 |
| TimeFlow | 0.41 | 0.85 | 3.19 | 2.177 | 1.17 | 0.20 | 0.65 | 8.40 | 173.12 | - | 3.45 |
| TimeGAN | 0.88 | 0.87 | 20.32 | 2.00 | 2.26 | 3.88 | 0.70 | 4.73 | 6.63 | 1.30 | 3.97 |
| GTGAN | 0.70 | 2.55 | 26.60 | 3.23 | 25.53 | 3.39 | 2.82 | 16.23 | 3.23 | 2.24 | 10.01 |
| KoVAE | 0.48 | 1.17 | 6.78 | 1.72 | 8.82 | 1.17 | 0.80 | 2.46 | 2.89 | 3.85 | 6.54 |
| TimeVQVAE | 2.45 | 6.05 | 8.40 | 0.74 | 5.06 | 4.20 | 2.93 | 8.17 | 3.77 | 4.62 | 3.98 |
| LS4 | 5.85 | 10.97 | 23.47 | 3.47 | 15.81 | 24.21 | 31.81 | 14.45 | 8.15 | 11.34 | 24.67 |
| VAE | 4.41 | 7.16 | 35.65 | 1.67 | 28.16 | 3.42 | 2.62 | 15.70 | 7.66 | 6.88 | 10.02 |

Table 3: Contextual-FID results on 11 time series datasets. The lower the better. **Bold** and underline indicate the best and second best performance, respectively. The symbol "-" indicates that the method suffers from unstable training and significantly inferior performance.

**Evaluation Metrics.** Following Ang et al. (2023), we evaluate generation quality using the following three metrics on the test set: (i) Contextual-FID (C-FID) (Zhou et al., 2022), which measures how well the synthetic time series align with the local context of the original data; (ii) Discriminative Score (DS), and (iii) Predictive Score (PS). The use of DS and PS follows (Yuan & Qiao, 2024). For DS, a 2-layer LSTM is trained to classify sequences as "real" (original) or "not real" (generated), with the classification error measuring dataset similarity. For PS, a 2-layer LSTM is trained on the generated data to predict next-step temporal vectors, and its mean prediction error on the original dataset reflects how well predictive patterns are preserved. However, we consider DS and PS as secondary metrics due to their sensitivities to model setup and dataset size. Besides, we also provide visualizations using t-SNE and distribution plots to compare the distributions of the original and generated time series.

**Implementation Details.** We train the model using Adam with a learning rate of $10^{-3}$, batch size of 128, and early stopping for up to 100 epochs. Both the latent and data diffusion models in L2D-Diff are trained with $K = 100$ diffusion steps under a linear variance schedule (Rasul et al., 2021), with $\beta_1 = 10^{-4}$ and $\beta_K = 10^{-1}$. The CNN of TS2Vec (Yue et al., 2022) is pre-trained as our encoder, with a default latent dimension of 8 for high-level representation learning. The decoder utilizes a three-layer convolutional network. The masked ratio is set to be 50% as in (Dong et al., 2023). Implementation details of the denoising network are in Appendix F. Experiments are run on an Nvidia RTX A6000 GPU with 48GB of memory. All baselines are evaluated using their official implementations.

## 5.2 MAIN RESULTS

Results on Contextual-FID are shown in Table 3. To validate the statistical significance of method rankings, we employ the Friedman test (Friedman, 1937) and Conover's post-hoc test (Conover & Iman, 1979). Figure 2 shows the average rankings and the corresponding critical differences (CD) for each method. The average ranking reflects the overall performance of each method (the lower the better). The CD value indicates the smallest difference in rankings that is statistically significant, as determined by the post-hoc test.

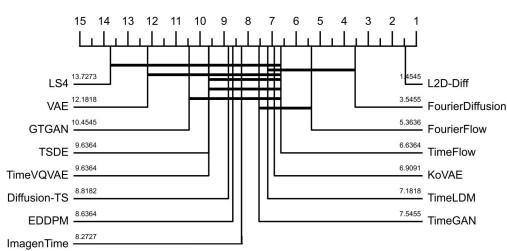

Figure 2: Critical difference diagram of TSG methods. The lower the better.

As can be seen, the proposed L2D-Diff achieves superior overall performance with an average rank of 1.45, significantly outperforming all the baselines. L2D-Diff is simple yet effective. As a cas-

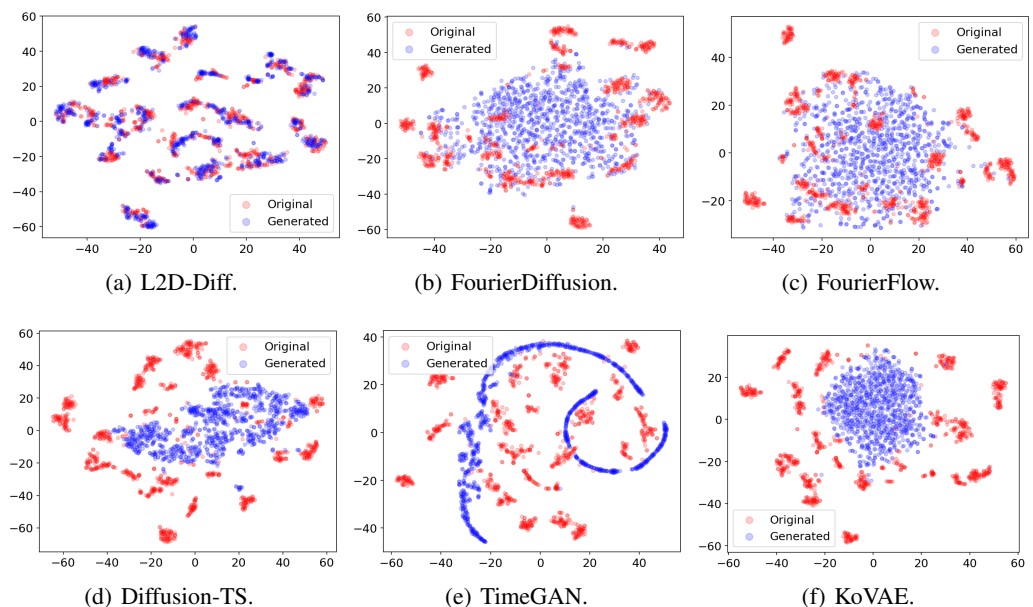

Figure 3: 2D t-SNE embeddings of data (not representations) generated vs the real data of *Character Trajectories* with multiple modes.

caded diffusion model, it bridges the latent and data diffusion processes, transforming an unconditional time series generation problem into a conditional one. Specifically, the latent diffusion model first captures the high-level representation distribution, which then guides the data-space denoising process. This approach not only reduces complexity but also ensures efficient and robust generation of time series, particularly when dealing with complex or multimodal distributions.

Among the baselines, FourierDiffusion (Crabbé et al., 2024) ranks second with an average rank of 3.55, followed by FourierFlow (Alaa et al., 2021) with an average rank of 5.36. Fourier Diffusion introduces the concept of mirrored Brownian motion and performs data generation in the frequency domain, while Fourier Flow leverages discrete Fourier transform to convert time series into fixed-length spectral representations and applies a data-dependent spectral filter to these transformed series. Moreover, TSDE underperforms due to its lack of high-level latent guidance. Results for Discriminative Score (DS) and Predictive Score (PS) are presented in Table 6 of Appendix D.

## 5.3 VISUALIZATION RESULTS

Figure 3 shows the 2D t-SNE embeddings produced by the proposed L2D-Diff and several popular baselines on the *Character Trajectories* dataset. As can be seen, this dataset is challenging due to its complex multi-modal distribution across 20 classes, limited training samples, and the need to model numerous modes effectively. The proposed L2D-Diff overcomes these difficulties by leveraging its latent-to-data dual-space framework, capturing global structures while preserving local fidelity, and generating time series that replicate the original data's features and patterns with high accuracy. In contrast, competing baselines tend to capture the overall distribution center while failing to adequately represent its diversity. More visualization results are in Appendix E.

## 5.4 ABLATION STUDIES

In this section, we perform an ablation study on the effectiveness of latent-space and data-space diffusion using the *Stock* and *Character Trajectories* datasets. We compare the proposed L2D-Diff with two variants: (i) Latent-space only, which uses only latent-space diffusion by removing the data-space branch and decoding with a pretrained decoder, and (ii) Data-space only, which uses only the data-space diffusion by replacing the latent condition **c** with zeros. More ablation results on the effects of the latent dimension and conditioning network are in Appendices G and H, respectively.

|  | | Stock | | Character Trajectories | | |
|---|---|---|---|---|---|---|
|  | C-FID | DS | PS | C-FID | DS | PS |
| **L2D-Diff** (full) | **0.310** | **0.048** | **0.041** | **0.284** | **0.179** | **0.333** |
| Latent-space only | 3.682 | 0.204 | 0.089 | 1.829 | 0.355 | 0.353 |
| Data-space only | 0.385 | 0.049 | 0.052 | 2.368 | 0.380 | 0.369 |

Table 4: Effectiveness of the latent-space-only and data-space-only variants.

Table 4 shows the C-FID, DS and PS results. As can be seen, on *Stock*, the data-space variant outperforms the latent-space one. We speculate that it is because this time series is short ($L = 24$) and the distribution is simple (as shown in Figure 5(a) of Appendix E). On the other hand, for the more difficult *Character Trajectories* dataset, the latent-space variant performs better, indicating the effectiveness of global semantics. In both cases, L2D-Diff consistently outperforms the two variants. This demonstrates that combining latent-space and data-space diffusion is crucial for achieving both global coherence and local fidelity in time series generation.

## 5.5 EFFICIENCY

In this section, we evaluate the efficiency of L2D-Diff against four representative diffusion models: (i) Diffusion-TS, a recent popular data-space model; (ii) TimeLDM, a latent-space diffusion model; (iii) FourierDiffusion, the most competitive baseline.

|  | type | training (ms/sample) | inference (ms/sample) | # of trainable parameters |
|---|---|---|---|---|
| **L2D-Diff** | data + latent | 0.52 | 3.47 | 2.2M |
| Diffusion-TS | data | 14.28 | 5.10 | 25M |
| TSDE | data | 2.10 | 5.05 | 1.3M |
| TimeLDM | latent | 0.51 | 4.85 | 1.9M |
| EDDPM | latent | 0.51 | 3.80 | 1.9M |
| FourierDiffusion | frequency | 0.36 | 9.66 | 1.6M |
| ImagenTime | frequency | 1.22 | 2.82 | 1.1M |

Table 5: Training & inference time, and number of trainable parameters on the *Character Trajectories*.

Table 5 compares their training time, inference time, and number of trainable parameters on the *Character Trajectories* dataset. Results on additional datasets are in Appendix I. As can be seen, compared to existing time series diffusion models, the proposed L2D-Diff is efficient because its latent-space diffusion process is learned in a low-dimensional latent space ($d \ll D \times L$, $d = 8$ by default). By leveraging convolution layers and the acceleration technique DPM-Solver (Lu et al., 2022), L2D-Diff achieves a significant reduction in computational costs without compromising generation quality. This demonstrates its efficiency in handling complex multi-modal time series data while maintaining a lightweight model design.

## 6 CONCLUSION

We proposed L2D-Diff, a simple yet efficient dual-space diffusion framework for high-fidelity time series generation. By integrating dual-space diffusion processes, L2D-Diff learns representation distribution in a compressed latent space and generation time series in the data space under latent guidance. This streamlined design effectively balances simplicity, fidelity, and efficiency, achieving state-of-the-art performance across a wide range of datasets. Extensive experiments validate L2D-Diff's ability to generate realistic and coherent time series while preserving multi-modal distribution structures. In summary, L2D-Diff exemplifies how a simple yet efficient approach can address the challenges of unconditional time series generation. We hope this work inspires the development of more lightweight and scalable diffusion models. The official implementation of L2D-Diff is publicly available at *link* to support reproducibility and further exploration.

## ACKNOWLEDGMENTS

This research was supported in part by the National Natural Science Foundation of China (Grants 62221005, 62576056) and the Research Grants Council of the Hong Kong Special Administrative Region (Grants 16200021, 16202523 and HKU C7004-22G).

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

# A    RELATED WORKS ON TIME SERIES GENERATION

Generative models aim to learn intricate patterns and temporal dependencies in time series datasets, enabling the generation of new data that reflects the statistical properties of the original dataset (Ang et al., 2023). In addition to the recent time series diffusion models discussed in Section 2, this section reviews three major categories of classic generative models for time series generation: generative adversarial networks (GANs), variational autoencoders (VAEs), and flow-based generative models.

Generative adversarial networks (GANs) (Goodfellow et al., 2014) consist of a generator and a discriminator, trained through a two-player minimax game. The generator takes random noise as input and learns to produce synthetic data that is indistinguishable from the real data, while the discriminator is tasked with classifying real and generated samples. In the context of time series generation, GANs have been enhanced by incorporating specialized generator architectures, such as LSTMs or Transformers, to improve the modeling of temporal dependencies (Esteban et al., 2017; Li et al., 2022; Mogren, 2016; Pei et al., 2021; Yoon et al., 2019). Additionally, various strategies have been proposed to improve the training process, including novel loss functions, extra discriminators, classification layers, and data augmentation techniques, which aim to achieve better temporal alignment and enhance performance (Ni et al., 2021; Jeha et al., 2022; Seyfi et al., 2022; Wang et al., 2023). Despite their effectiveness, GAN-based models are often challenging to train due to instability in the adversarial process and are computationally expensive, requiring significant resources and time (Jeon et al., 2022; Ang et al., 2023).

Variational autoencoders (VAEs) (Kingma & Welling, 2014) offer an alternative approach by minimizing a combination of reconstruction loss and the divergence between the learned latent distribution and a prior standard Gaussian distribution. VAEs effectively leverage variational inference to capture complex temporal relationships in time series data (Desai et al., 2021; Lee et al., 2023; Li et al., 2023). A notable example is TimeVQVAE (Lee et al., 2023), which integrates vector quantization (Van Den Oord et al., 2017) to preserve both the general shape and fine-grained details of time series. Another recent work, LS4 (Zhou et al., 2023), models latent space evolution using a state space ordinary differential equation (ODE) and is trained with standard sequence VAE objectives.

In addition to GANs and VAEs, flow-based generative models have also been extended to time series generation (Dinh et al., 2024; Alaa et al., 2021). Unlike GANs and VAEs, flow-based models directly model the probability density function of time series, avoiding the computational challenges of sampling from latent representation distributions. For instance, Fourier Flow (Alaa et al., 2021) employs a novel class of normalizing flows combined with discrete Fourier transforms (DFT) to convert variable-length time series with arbitrary sampling periods into fixed-length spectral representations. A data-dependent spectral filter is then applied to refine the frequency-transformed time series, enabling explicit likelihood estimation.

Most recently, Sikder et al. (2025) developed a Transformer-based diffusion model called TransFusion for long-sequence generation. However, our research focus differs; we emphasize how to better facilitate time series generation with multiple modes using a novel latent-to-data cascaded structure.

# B    THEORETICAL UNDERSTANDING

In this section, we provide theoretical justifications of the proposed latent-to-data method using the Information Bottleneck (IB) principle.

## B.1    ENTROPY DECOMPOSITION AND CASCADED GENERATION

First, we decompose the time series generation process using the chain rule of entropy. Let $X$ be the observed time series and $Z$ be the latent representation. The total entropy of the data $H(X)$ can be decomposed into two components:

$$H(X) = \underbrace{I(X;Z)}_{\text{Global Information}} + \underbrace{H(X|Z)}_{\text{Local Detail / Residual Uncertainty}}. \tag{15}$$

This decomposition provides a formal basis for our latent-to-data diffusion architecture, assigning distinct and complementary roles to each model.

**Stage 1 (Latent Diffusion $P(Z)$): Modeling $I(X; Z)$.** The first stage learns the distribution of the latent code $Z$. Since $Z$ is a compressed representation, $I(X; Z)$ corresponds to the *global semantic structure*. By modeling $P(Z)$, the first diffusion model captures the high-level semantics of the data.

**Stage 2 (Data Diffusion $P(X|Z)$): Modeling $H(X|Z)$.** The second stage models the conditional distribution of $X$ given $Z$. This term represents the *local details* or residual uncertainty not captured by $Z$. Crucially, we show that this conditional distribution often has *high entropy* and is *multi-modal*. Our framework employs a diffusion model here precisely because diffusion models excel at sampling from such complex distributions, effectively "synthesizing" the missing high-frequency details that deterministic decoders would average out. The marginal distribution is thus recovered via the integral:

$$P_\theta(X) = \int P_\theta(X|Z)P(Z)\, dZ \approx P_{\text{data}}(X).$$

This proves that the cascaded approach is theoretically capable of recovering the true data distribution, provided both conditional and latent models are well-learned.

### B.2 INFORMATION BOTTLENECK TRADE-OFF AND IMPACT OF LATENT DIMENSION

Next we provide the theoretical understanding of the trade-off imposed by the latent dimension $d$ through the lens of the Information Bottleneck principle.

**The Bottleneck Constraint:** A compact latent dimension $d$ imposes a capacity constraint $C(d)$ on the mutual information, i.e., $I(X; Z) \leq C(d)$. As $d$ decreases, $C(d)$ decreases, forcing $Z$ to discard more information. Using the entropy decomposition in (15), a reduction in $I(X; Z)$ (due to compression) necessarily increases the conditional entropy $H(X|Z)$:

$$H(X|Z) = H(X) - I(X; Z) \geq H(X) - C(d).$$

This inequality quantifies the "burden shift":

**Small $d$ (High Compression):** $Z$ with a much smaller $d$ retains only very coarse global semantics. The burden of modeling data complexity shifts to the conditional model $P(X|Z)$, which must now model a highly uncertain, multi-modal distribution. Our use of a conditional diffusion model is critical here, as it prevents the "blurriness" or information loss typical of deterministic decoders by effectively sampling from this high-entropy distribution.

**Large $d$ (Low Compression):** In this case, $Z$ retains more high-level information, reducing $H(X|Z)$. However, modeling $P(Z)$ with larger $d$ becomes more challenging as the latent space becomes high-dimensional and complex, potentially leading to overfitting or difficulty in learning the global manifold.

The theoretical insight emphasizes a crucial mechanism: as the small latent representation $Z$ compresses (leading to a decrease in $I(X; Z)$), the onus of capturing data complexity shifts to the conditional term $P_\theta(X|Z)$. Our framework excels in this regard, as the data-space diffusion model adeptly captures the heightened conditional entropy, thereby mitigating the information loss typically associated with deterministic decoders.

Moreover, even with a rich latent space, our cascaded structure provides distinct advantages over single-stage models, particularly for multi-modal time series generation. Directly modeling the marginal $P(X)$ necessitates a model that navigates diverse semantics and complex local variations, often resulting in mode mixing. By decoupling the generation process, our framework delegates *global semantic modeling*—represented by $I(X; Z)$—to the latent model, facilitating clearer and more learnable abstract structures. This "divide-and-conquer" strategy ensures generated time series exhibit coherent high-level semantic structures and realistic local details.

## C  TRAINING AND SAMPLING ALGORITHMS

Algorithms 1 and 2 show the pseudocodes for the training and sampling processes, respectively.

---

**Algorithm 1** Training of L2D-Diff.

---

**Require:** Training dataset $\mathcal{T}$, noise schedules $\{\beta_t\}_{t=1}^T$.
**Ensure:** Trained latent-space denoising network $\mathbf{r}_\phi$, data-space denoising network $\mathbf{x}_\theta$, and conditioning network $\mathcal{F}$.
  **while** not converged **do**
    $s \sim \text{Uniform}(\{1, 2, \ldots, S\})$, $k \sim \text{Uniform}(\{1, 2, \ldots, K\})$;
    Sample $\mathbf{x} \sim \mathcal{T}$;
    Generate latent embedding $\mathbf{r} = \mathbf{E}(\mathbf{x})$;
    Generate noised latent $\mathbf{r}^s = \sqrt{\bar{\alpha}_s}\mathbf{r} + \sqrt{1 - \bar{\alpha}_s}\epsilon$, where $\epsilon \sim \mathcal{N}(\mathbf{0}, \mathbf{I})$ and $\epsilon \in \mathbb{R}^d$;
    Compute latent denoising loss $\mathcal{L}_{\text{latent}}$ in Equation ( 11).
    Obtain latent-to-data condition $\mathbf{c} = \mathbf{r}$;
    Generate noised data $\mathbf{x}^k = \sqrt{\bar{\alpha}_k}\mathbf{x} + \sqrt{1 - \bar{\alpha}_k}\epsilon$, where $\epsilon \sim \mathcal{N}(\mathbf{0}, \mathbf{I})$ and $\epsilon \in \mathbb{R}^{D \times L}$;
    Compute data denoising loss $\mathcal{L}_{\text{data}}$ in Equation (12);
    Update $\phi, \theta$ via $\nabla_{\phi,\theta}(\mathcal{L}_{\text{latent}} + \lambda \cdot \mathcal{L}_{\text{data}})$ ($\lambda$ is 1 by default);
  **end while**

---

**Algorithm 2** Sampling of L2D-Diff.

---

**Require:** Trained denoising models $\mathbf{r}_\phi$, $\mathbf{x}_\theta$ and conditioning network $\mathcal{F}$.
**Ensure:** Generated sample $\hat{\mathbf{x}}^0$.
  **Latent-space generation:**
  Sample $\hat{\mathbf{r}}^S \sim \mathcal{N}(0, I)$.
  **for** $s = S$ **downto** 1 **do**
    Denoise latent: $\hat{\mathbf{r}}^{s-1} = \mathbf{r}_\phi(\mathbf{r}^s, s)$ by Equation (13).
  **end for**
  **Data-space refinement:**
  Sample $\hat{\mathbf{x}}^K \sim \mathcal{N}(0, I)$.
  Compute the condition $\mathbf{c} = \mathcal{F}(\hat{\mathbf{r}}^0)$.
  **for** $t = T$ **downto** 1 **do**
    Denoise data: $\hat{\mathbf{x}}^{k-1} = \mathbf{x}_\theta(\mathbf{x}^k, k, \mathbf{c})$ by Equation (14).
  **end for**
  **return** $\hat{\mathbf{x}}^0$

---

## D  DISCRIMINATIVE SCORE AND PREDICTIVE SCORE

From each category of the baselines, we use the top one from Figure 2. Table 6 reports the evaluation results on discriminative and predictive scores. As can be seen, L2D-Diff consistently achieves the best overall performance, outperforming all the baselines. This demonstrates the effectiveness of the L2D dual-space framework in tackling the key challenge of TSG: capturing global structures in the latent space while preserving local fidelity in the data space.

For several datasets (*Riverflow*, *Two Patterns*, *ECG5000*, *Medical Images*, and *Atrial Fibrillatio*), all baselines achieve a DS value of 0. A DS value of 0 indicates that the discriminator cannot reliably distinguish real from generated sequences, meaning the classification error is close to random guessing. This suggests that under the specific LSTM-based discriminator setup (Section 5.1), the generated samples are statistically similar to the real data. This phenomenon does not necessarily imply that these datasets are overly simple or that the generative performance is perfect. Since DS is highly sensitive to the discriminator architecture, training protocol, and dataset size, a zero value may also reflect limited discriminative power under the given evaluation configuration. That is why we treat DS as a secondary metric and interpret it alongside the other quantitative measures and qualitative visualizations.

|    |                  | Stock | Energy | ETTh | Riverflow | Two Patterns | ECG5000 | Medical Images | Arabic Digits | Atrial Fibrillation | Japanese Vowels | Character Trajectories | Win/Tie |
|----|------------------|-------|--------|------|-----------|--------------|---------|----------------|---------------|---------------------|-----------------|------------------------|---------|
| DS | **L2D-Diff**     | 0.048 | **0.166** | **0.009** | **0.000** | **0.000** | **0.000** | **0.000** | 0.298 | **0.000** | **0.027** | 0.179 | 8 |
|    | Diffusion-TS     | **0.007** | 0.420 | 0.101 | **0.000** | **0.000** | **0.000** | **0.000** | 0.368 | **0.000** | 0.324 | 0.385 | 6 |
|    | TimeLDM          | 0.493 | 0.495 | 0.489 | **0.000** | **0.000** | **0.000** | **0.000** | 0.475 | **0.000** | 0.324 | 0.323 | 5 |
|    | FourierDiffusion | 0.174 | 0.316 | 0.153 | **0.000** | **0.000** | **0.000** | **0.000** | 0.451 | **0.000** | 0.203 | 0.321 | 5 |
|    | FourierFlow      | 0.221 | 0.394 | 0.381 | **0.000** | **0.000** | **0.000** | **0.000** | 0.481 | **0.000** | 0.216 | 0.253 | 5 |
|    | TimeGAN          | 0.218 | 0.496 | 0.494 | **0.000** | **0.000** | **0.000** | **0.000** | 0.492 | **0.000** | 0.142 | **0.164** | 6 |
|    | KoVAE            | 0.054 | 0.214 | 0.089 | **0.000** | **0.000** | **0.000** | **0.000** | **0.220** | **0.000** | 0.392 | 0.297 | 6 |
| PS | **L2D-Diff**     | **0.041** | **0.251** | **0.654** | 0.049 | **0.754** | 0.556 | 0.623 | **0.333** | **0.539** | 0.331 | **0.333** | 8 |
|    | Diffusion-TS     | 0.048 | 0.269 | 0.905 | 0.049 | 0.755 | 0.599 | 0.801 | 0.338 | **0.539** | 0.365 | 0.368 | 1 |
|    | TimeLDM          | 0.078 | 0.278 | 0.889 | 0.064 | 0.755 | 0.671 | 0.631 | 0.365 | 1.081 | 0.355 | 0.347 | 0 |
|    | FourierDiffusion | 0.051 | 0.252 | 0.780 | 0.050 | 0.755 | **0.551** | 0.626 | 0.338 | 0.540 | 0.340 | 0.355 | 1 |
|    | FourierFlow      | 0.108 | 0.269 | 0.823 | 0.064 | 0.755 | 0.554 | 0.654 | 0.343 | 0.540 | 0.338 | 0.355 | 0 |
|    | TimeGAN          | 0.045 | 0.293 | 0.889 | 0.038 | 0.754 | 0.611 | 0.645 | 0.343 | 0.707 | 0.361 | 0.347 | 2 |
|    | KoVAE            | 0.047 | 0.257 | 0.782 | 0.038 | 0.754 | 0.554 | **0.619** | 0.341 | 0.542 | 0.367 | 0.340 | 3 |

Table 6: Results on the time series datasets. Top: Discriminative Score (DS); Bottom: Predictive Score (PS). The lower the better. The last column counts the number of wins or ties for each method.

# E VISUALIZATION

Figure 4 shows the distribution plot results. The distribution plot illuminates the difference between the input and generated time series in terms of density, spread, and central tendency to show how the generated time series closely mirrors the original's statistics (Ang et al., 2023). The proposed L2D-Diff demonstrates exceptional performance, consistently producing synthetic data with a distribution that closely mirrors the original, regardless of the complexity of the underlying data.

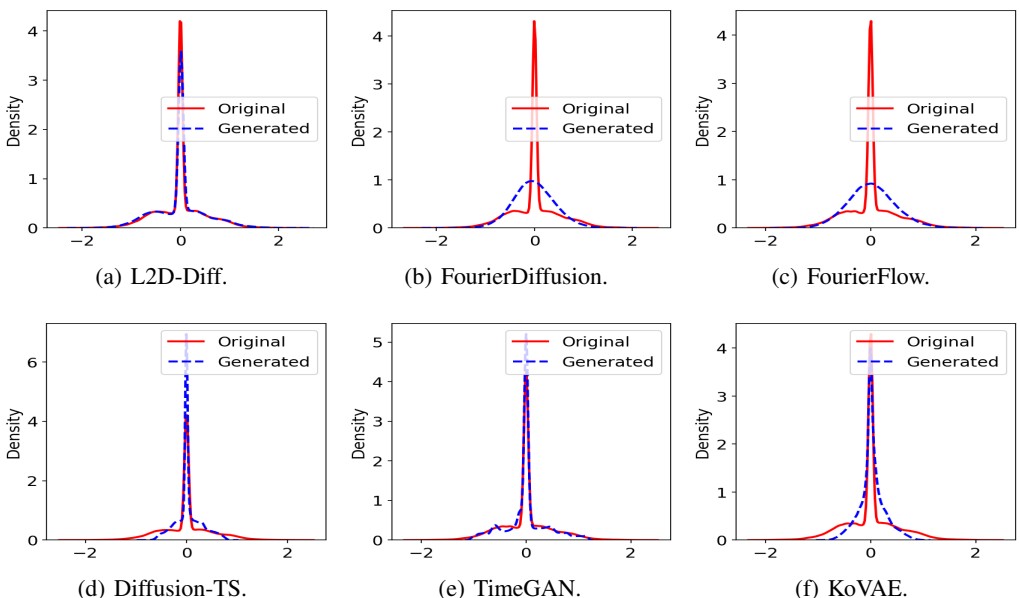

Figure 4: Distribution plot results on *Character Trajectories*.

Figure 5 and Figure 6 show the t-SNE results for the other datasets. As can be seen, the proposed L2D-Diff consistently delivers the best performance in producing data distributions that closely align with the true multimodal distribution.

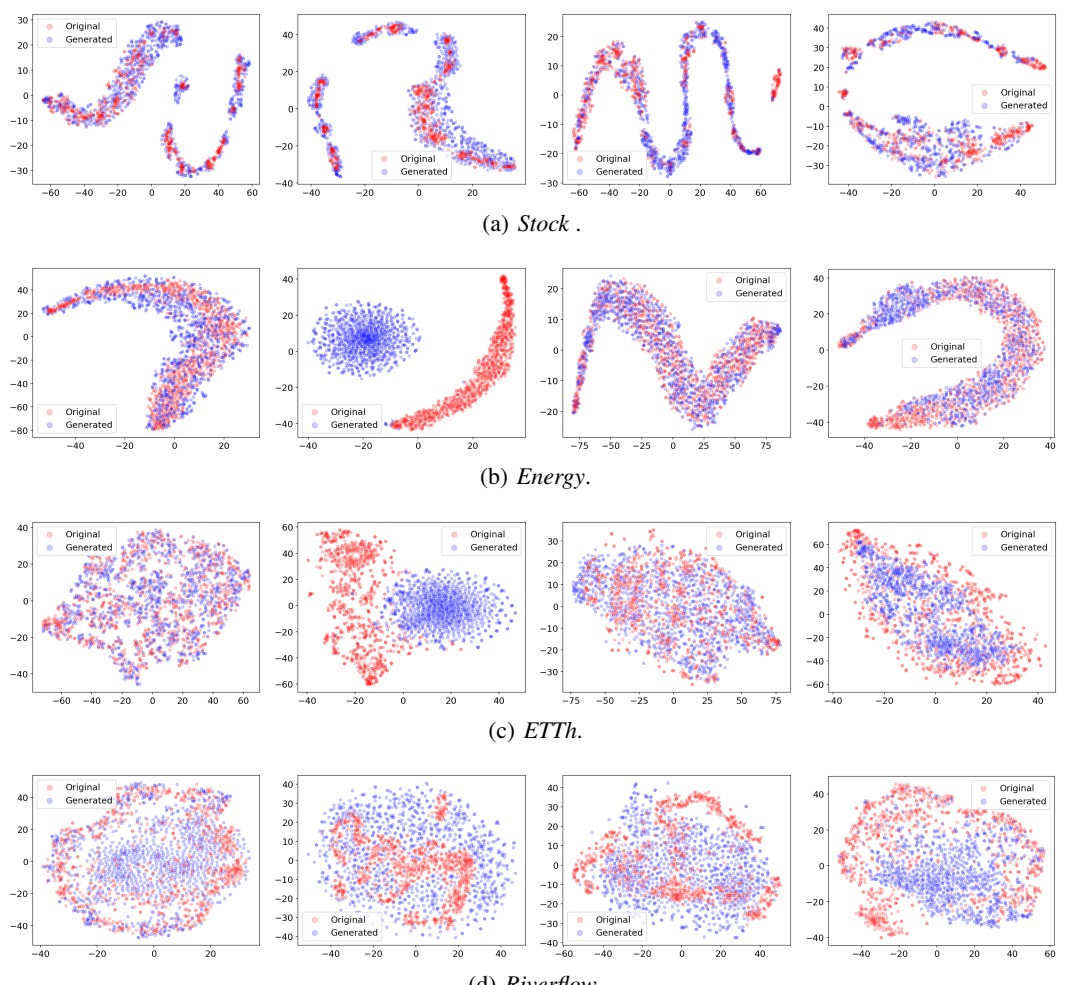

(a) *Stock* .

(b) *Energy*.

(c) *ETTh*.

(d) *Riverflow*.

Figure 5: 2D t-SNE embeddings of data (not representations) generated vs the real data. Column 1: L2D-Diff; Column 2: FourierDiffusion; Column 3: FourierFlow; Column 4: Diffusion-TS.

## F IMPLEMENTATION DETAILS OF THE DENOISING NETWORK

### F.1 DIFFUSION STEP'S EMBEDDINGS

For each diffusion step $k$, its $d'$-dimensional embedding $\mathbf{p}^k$ is computed using two fully connected (FC) layers, as in (Rasul et al., 2021; Tashiro et al., 2021; Kong et al., 2020):

$$\mathbf{p}^k = \mathrm{SiLU}(\mathrm{FC}(\mathrm{SiLU}(\mathrm{FC}(k_{\texttt{embedding}})))), \tag{16}$$

where SiLU is the sigmoid-weighted linear unit activation function (Elfwing et al., 2018), and

$$k_{\texttt{embedding}} = \left[\sin(10^{\frac{0\times4}{w-1}}t), \ldots, \sin(10^{\frac{w\times4}{w-1}}t), \cos(10^{\frac{0\times4}{w-1}}t), \ldots, \cos(10^{\frac{w\times4}{w-1}}t)\right] \tag{17}$$

is the sinusoidal position embedding (Vaswani et al., 2017) with $w = \frac{d'}{2}$. By default, $d'$ is set to 128.

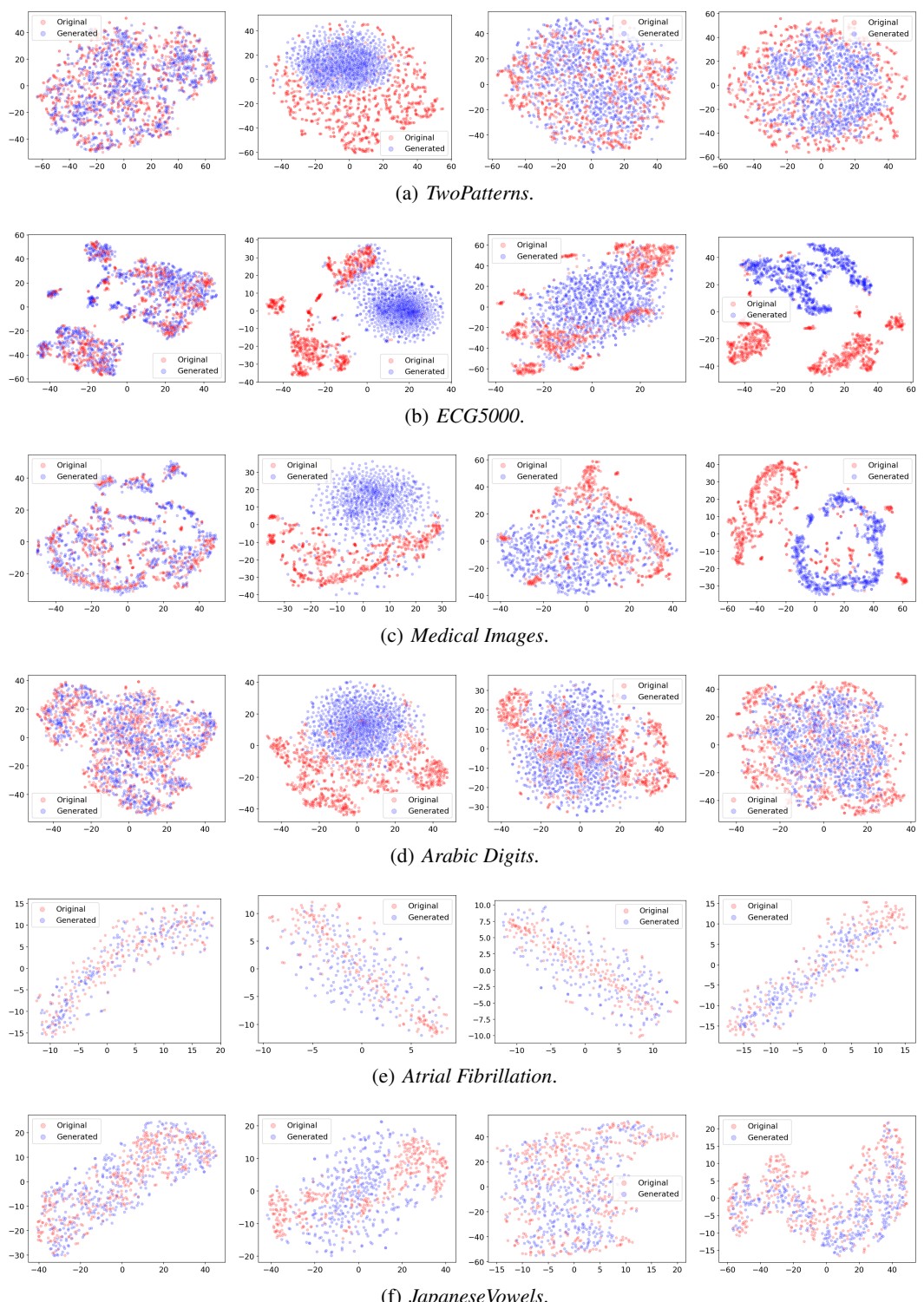

(a) *TwoPatterns*.

(b) *ECG5000*.

(c) *Medical Images*.

(d) *Arabic Digits*.

(e) *Atrial Fibrillation*.

(f) *JapaneseVowels*.

Figure 6: 2D t-SNE embeddings of data generated vs the real data. Column 1: L2D-Diff; Column 2: FourierDiffusion; Column 3: FourierFlow; Column 4: Diffusion-TS.

## F.2 Denoising Network Workflow

Take the data denoising network $\mathbf{x}_\theta$ as an example. The input $\mathbf{x}^k \in \mathbb{R}^{D \times L}$ is first mapped to the embedding $\bar{\mathbf{z}}^k \in \mathbb{R}^{d' \times L}$ by an input projection block consisting of several convolutional layers.

The embedding $\bar{\mathbf{z}}^k$, along with the $d'$-dimensional diffusion step embedding $\mathbf{p}^k$ (from Equation (16)), is then passed to an encoder (a convolutional network) to produce the representation $\mathbf{z}^k \in \mathbb{R}^{d'' \times L}$. Next, $\mathbf{z}^k$ is concatenated with $\mathbf{z}^c$ (which has a size of $d_c \times L$ after being upsampled to length $L$ by the conditioning network $\mathcal{F}$, and $d_c$ is the number of channels in $\mathbf{z}^c$) along the variable dimension, forming a tensor of size $(d_c + d'') \times L$. This concatenated tensor is then passed to a decoder, also implemented as a convolutional network, which outputs the denoised estimation $\mathbf{x}_\theta(\mathbf{x}^k, k, \mathbf{c})$.

In the latent-space denoising network $\mathbf{r}_\phi$, the corresponding representation $\mathbf{z}^s$ is directly fed into the decoder, which outputs the final denoised estimation $\mathbf{r}_\phi(\mathbf{r}^s, s)$.

## F.3 Network Implementation

The conditioning network and denoising network's encoder/decoder are built by stacking a number of convolutional blocks. The default configuration of each convolutional block is shown in Table 7.

| layer | operator | default parameters |
|---|---|---|
| 1 | Conv1d | in channel=256, out channel=256, kernel size=3, stride=1, padding=1 |
| 2 | BatchNorm1d | number of features=256 |
| 3 | LeakyReLU | negative slope=0.1 |
| 4 | Dropout | dropout rate=0.1 |

Table 7: Configuration of the convolutional block.

## G Effects of the Latent Dimension

As L2D-Diff bridges the diffusion process in the latent and data spaces, the dimension of the latent space plays a crucial role. We study its effects by varying the dimension sizes in {4, 8, 32, 64, 128}. As shown in Table 8, smaller latent dimensions, such as 8 or 32, yield promising results. This is reasonable since smaller latent spaces compress data and extract the most informative representations, effectively capturing the essential structures of the time series. In contrast, higher dimensions, such as 64 or 128, tend to increase training complexity and may lead to overfitting, as they retain more redundant or less informative details.

| $d$ | *Stock* | | | *Character Trajectories* | | |
|---|---|---|---|---|---|---|
| | C-FID | DS | PS | C-FID | DS | PS |
| 4 | 0.334 | 0.052 | 0.048 | 0.308 | 0.169 | 0.363 |
| 8 | **0.310** | **0.048** | **0.041** | **0.284** | 0.171 | **0.333** |
| 32 | 0.339 | 0.078 | 0.047 | 0.304 | **0.165** | 0.342 |
| 64 | 1.121 | 0.095 | 0.049 | 2.614 | 0.175 | 0.372 |
| 128 | 0.366 | 0.071 | 0.059 | 2.947 | 0.243 | 0.382 |

Table 8: Varying the latent dimension $d$.

# H ABLATION STUDIES ON CONDITIONING NETWORK $\mathcal{F}$

In this section, we present several ablation studies on the conditioning network $\mathcal{F}$: (i) the impact of varying the depth of $\mathcal{F}$; (ii) exploring the use of MLP as an alternative for $\mathcal{F}$; and (iii) examining different conditioning strategies.

## H.1 VARYING THE DEPTH OF CONDITIONING NETWORK $\mathcal{F}$

To investigate the impact of the depth of $\mathcal{F}$, Table H.1 varies the depth to {1, 3, 5, 10, 20}. The results indicate that the depth of $\mathcal{F}$ has a relatively stable range; it should neither be too shallow nor excessively deep. A single layer may result in inadequate learning of conditions, while an overly deep architecture can complicate the overall model structure and ultimately leads to overfitting. Empirically, in our experiments, a depth of 5 layers shows promising performance with fewer parameters compared to a model with 10 layers.

| No. Layers | Stocks | Energy | Japanese Vowels | Character Trajectories |
|---|---|---|---|---|
| 1 | 0.59 | 0.71 | 1.39 | 0.58 |
| 3 | 0.34 | 0.62 | 0.58 | 0.42 |
| 5 | 0.31 | **0.53** | **0.46** | 0.28 |
| 10 | **0.30** | **0.53** | 0.51 | **0.24** |
| 20 | 0.75 | 0.87 | 0.85 | 0.61 |

Table 9: C-FID results of varying depth of $\mathcal{F}$.

## H.2 MLP OR CNN

Table H.2 explores the effect of replacing the CNN in $\mathcal{F}$ with an MLP. The results clearly show that the CNN consistently outperforms the MLP, indicating that convolutional structures are better suited for this task than MLPs in this context.

| | Stocks | Energy | Japanese Vowels | Character Trajectories |
|---|---|---|---|---|
| CNN | **0.31** | **0.53** | **0.46** | **0.28** |
| MLP | 0.48 | 0.61 | 0.57 | 0.43 |

Table 10: Types of $\mathcal{F}$.

## H.3 DIFFERENT CONDITIONING STRATEGIES

Finally, we analyze various conditioning strategies, including concatenation, cross-attention, FiLM, and direct addition. Results are shown in Table 11. As can be seen, concatenation consistently outperforms the other three alternatives. While FiLM is also a viable option, it exhibits slightly unstable performance and incurs higher inference costs. In contrast, cross-attention is ineffective for bridging the latent space with the data space.

| C-FID | Stocks | Energy | Japanese Vowels | Character Trajectories | Training | Inference |
|---|---|---|---|---|---|---|
| Concatenate | **0.31** | **0.53** | **0.46** | **0.28** | **0.52** | **3.47** |
| Cross-attention | 0.39 | 0.68 | 0.49 | 0.61 | 0.71 | 4.13 |
| FiLM | 0.32 | 0.55 | 0.48 | 0.37 | 0.54 | 3.89 |
| Addition | 0.55 | 0.81 | 0.72 | 0.97 | 0.49 | 3.10 |

Table 11: Effects of Conditioning Modules. Training and inference times are measured in milliseconds per sample.

# I TRAINIG & INFERENCE TIME

Table 12 shows the training time, inference time, and number of trainable parameters on the *Energy*, *Stock*, and *Character Trajectories* datasets. As can be seen, L2D-Diff consistently ranks among the most efficient models, achieving the comparable results in terms of parameter count and computational cost.

| models | type | *Energy* | | *Stock* | | *Character Trajectories* | |
|---|---|---|---|---|---|---|---|
| | | training (ms/sample) | inference (ms/sample) | training (ms/sample) | inference (ms/sample) | training (ms/sample) | inference (ms/sample) |
| **L2D-Diff** | data + latent | 0.32 | 1.28 | 0.44 | 1.95 | 0.52 | 3.47 |
| Diffusion-TS | data | 1.30 | 2.14 | 1.31 | 3.65 | 14.28 | 5.10 |
| TSDE | data | 0.37 | 6.55 | 0.83 | 8.63 | 2.10 | 5.05 |
| TimeLDM | latent | 0.28 | 2.11 | 0.35 | 3.12 | 0.51 | 4.85 |
| EDDPM | latent | 0.26 | 1.71 | 0.39 | 2.14 | 0.51 | 3.80 |
| FourierDiffusion | frequency | **0.19** | 6.10 | **0.28** | 8.11 | **0.36** | 9.66 |
| ImagenTime | fourier | 0.67 | **1.11** | 0.84 | **1.64** | 1.22 | **2.82** |

Table 12: Training and inference times (in milliseconds per sample) of various methods.

