# OpenReview forum: "Latent-to-Data Cascaded Diffusion Models for Unconditional Time Series Generation"
_ICLR.cc/2026/Conference — ICLR 2026 Poster_

### Official Review · Reviewer_AFRd · 2025-10-28

**Soundness:** 2
**Presentation:** 3
**Contribution:** 2
**Rating:** 4
**Confidence:** 4

**Summary:**

This paper proposes L2D-diff, a hybrid latent and data space diffusion model that aims to capture both high-level and local fidelity. The generation process first uses latent diffusion, then passes onto data space diffusion. Through empirical experiment, L2D-Diff is on par with previous SOTA models.

**Strengths:**

1. This paper investigates the hybrid diffusion models for time series generation, which is a new framework.
2. This paper conducted extensive experiments with newly added datasets.
3. This paper is well written and easy to follow.

**Weaknesses:**

1. The L2D-Diff consists of multiple modules. A comprehensive ablation study is needed to justify the hybrid framework.
2. While there are discussions of the training & inference, the discussion is limited only to a new dataset. The efficiency claim could be more compelling if the comparison can also be conducted over the sine, stock, and energy, which are more commonly compared.
3. The t-SNE visualization is only performed over one dataset. It would be ideal to include more datasets.

**Questions:**

Additional questions other than weaknesses.

1. In Table 7, the result for the previous SOTA methods, Diffusion-TS, seems to be very different from the original paper.
2. Newly proposed datasets, 5 out of 8 show 0 for DS score. Does that mean those datasets are too easy?

---

> ### Author Response · Authors · 2025-12-03
> **Response to Reviewer AFRd (Part 1)**
>
> We express our gratitude to the reviewer for providing constructive feedback on our paper. We have addressed the specific concerns raised by the reviewer as detailed below.
>
> **Q1: The L2D-Diff consists of multiple modules. A comprehensive ablation study is needed to justify the hybrid framework.**
>
> **A1:** As suggested, we present several ablation studies on our latent-to-data conditioning part:
> i) the impact of varying the depth of $\mathcal{F}$;
> ii) exploring the use of MLP as an alternative for $\mathcal{F}$;
> and iii) examining different conditioning strategies.
>
> To investigate the impact of the depth of $\mathcal{F}$, the table below shows how the C-FID varies with the number of encoders in {1, 3, 5, 10, 20}. The results indicate that the depth of the conditioning network $\mathcal{F}$ has a relatively stable range; it should neither be too shallow nor excessively deep. A single layer may result in inadequate learning of the conditions, while an overly deep architecture can complicate the overall model structure and ultimately lead to overfitting.
> Empirically, in our experiments, a depth of 5 layers shows promising performance with fewer parameters compared to a model with 10 layers.
>
> | No. Layers | Stocks        | Energy        | J.V.          | C.T.          |
> |------------|---------------|---------------|---------------|---------------|
> | depth=1    | 0.59          | 0.71          | 1.39          | 0.58          |
> | depth=3    | 0.34          | 0.62          | 0.58          | 0.42          |
> | depth=5    | 0.31        | **0.53**   | **0.46** | 0.28          |
> | depth=10   | **0.30** | **0.53** | 0.51          | **0.24** |
> | depth=20   | 0.75          | 0.87          | 0.85          | 0.61          |
>
> The table below explores the effect of replacing the CNN in $\mathcal{F}$ with an MLP. The C-FID results clearly show that the CNN consistently outperforms the MLP, indicating that convolutional structures are better suited for this task than multi-layer perceptrons.
>
> |               | Stocks | Energy | J.V. | C.T. |
> |---------------|--------|--------|------|------|
> | CNN           | 0.31   | 0.53   | 0.46 | 0.28 |
> | MLP           | 0.48   | 0.61   | 0.57 | 0.43 |
>
> Finally, we analyze various conditioning strategies, including concatenation, cross-attention, FiLM (Feature-wise Linear Modulation, which is originally designed for visual reasoning Ref [1]), and direct addition, as reported in the table on C-FID below.
>
> |                 | Stocks | Energy | J. V. | C. T. | Training time | Inference time |
> |-----------------|--------|--------|-------|-------|---------------|----------------|
> | Concatenate     | 0.31   | 0.53   | 0.46  | 0.28  | 0.52          | 3.47           |
> | Cross-attention | 0.39   | 0.68   | 0.49  | 0.61  | 0.71          | 4.13           |
> | FiLM            | 0.32   | 0.55   | 0.48  | 0.37  | 0.54          | 3.89           |
> | Addition        | 0.55   | 0.81   | 0.72  | 0.97  | 0.49          | 3.10           |
>
> The results show that the concatenation mechanism consistently outperforms the other three candidates. While FiLM is also a viable option, it exhibits slightly unstable performance and incurs higher inference costs. In contrast, cross-attention is ineffective for bridging the latent space with the data space.
>
> Ref [1]: AAAI'18, FiLM: Visual Reasoning with a General Conditioning Layer.

---

> ### Author Response · Authors · 2025-12-03
> **Response to Reviewer AFRd (Part 2)**
>
> **Q2: While there are discussions of the training \& inference, the discussion is limited only to a new dataset. The efficiency claim could be more compelling if the comparison can also be conducted over the sine, stock, and energy, which are more commonly compared.**
>
> **A2:** Thank you for your suggestions. As suggested, the training \& inference results (training and inference times are measured in milliseconds per sample.) on stock, and energy are added in the table below (has been added in Section K in the revised paper). As shown, the proposed L2D-Diff remains efficient.
>
> |                   |               | Energy |       |Stock |      |C.T. |      |
> |-------------------|---------------|--------|-------|--------|--------|------------------------|-------|
> |    model          | type         | training   |  inference  |   training   |  inference  |   training   |  inference  |
> | L2D-Diff | data + latent | 0.32   | 1.28  | 0.44                   | 1.95 | 0.52  | 3.47 |
> | mr-Diff           | data          | 0.89   | 3.54  | 0.94                   | 5.36 | 1.14  | 9.43 |
> | Diffusion-TS      | data          | 1.30   | 2.14  | 1.31                   | 3.65 | 14.28 | 5.10 |
> | TSDE              | data          | 0.37   | 6.55  | 0.83                   | 8.63 | 2.10  | 5.05 |
> | TimeLDM           | latent        | 0.28   | 2.11  | 0.35                   | 3.12 | 0.51  | 4.85 |
> | EDDPM             | latent        | 0.26   | 1.71  | 0.39                   | 2.14 | 0.51  | 3.80 |
> | FourierDiffusion  | frequency     | 0.19   | 6.10  | 0.28                   | 8.11 | 0.36  | 9.66 |
> | ImagenTime        | fourier       | 0.67   | 1.11  | 0.84                   | 1.64 | 1.22  | 2.82 |
>
> **Q3: The t-SNE visualization is only performed over one dataset. It would be ideal to include more datasets.**
>
> **A3:** Thank you for your suggestions. Additional t-SNE visualization results on all datasets have been added in the revised paper (lines 972-1078).
> As shown, the proposed L2D-Diff consistently delivers the best performance in producing data distributions that closely align with the true multimodal distribution.
>
> **Q4: In Table 7, the result for the previous SOTA methods, Diffusion-TS, seems to be very different from the original paper.**
>
> **A4:** Thank you for your careful reading. The difference in results arises from the training process used in Diffusion-TS, which involves all available data (including test data) during training and is tested on a subset of the samples seen during training. They implement a hyperparameter called “proportion,” where a training proportion of 1 indicates that all data is used, while a testing proportion of 0.9 is applied.
>
> In contrast, our setup differs from Diffusion-TS as we strictly evaluate on the test set, using a train/test split of 0.8 and 0.2. Our test samples were not included in the training process, which results in different outcomes compared to those reported in their work. This distinction can be verified in the open-source code available on their GitHub.
>
> **Q5: Newly proposed datasets, 5 out of 8 show 0 for DS score. Does that mean those datasets are too easy?**
>
> **A5:** No, it does not imply that the datasets are too easy.
>
> According to Ref. [1], the DS score uses a post-hoc time-series classification model with 2-layer LSTMs to distinguish between original and generated time series. Original series are labeled as "real," and generated ones as "synthetic." An RNN classifier is trained on these labels, and the classification error on a test set measures the fidelity of the generation model. The DS score reflects the absolute difference between the discriminator's output and 0.5; the closer this score is to 0.5, the better the model performs.
>
> It is important to refer to the C-FID results, which indicate that these datasets are, in fact, challenging. The situation you mentioned suggests that the discriminator used for the DS score, as outlined in Ref. [1] and commonly employed in previous works, may be somewhat inadequate for the newly proposed datasets. However, it performs satisfactorily for traditional stock and energy datasets. Thus, careful design is essential for meaningful evaluations.
>
> This is why we have chosen the relatively robust C-FID metric as our primary measure. We believe the reviewers may have overlooked this point, which we have clarified in the manuscript (see line 393 in the revised paper).
>
> Reference:
>
> [1] Jinsung Yoon, Daniel Jarrett, and Mihaela Van der Schaar. Time-series generative adversarial networks. Advances in neural information processing systems, 32, 2019.

---

### Official Review · Reviewer_8LUh · 2025-10-29

**Soundness:** 2
**Presentation:** 3
**Contribution:** 2
**Rating:** 4
**Confidence:** 3

**Summary:**

The paper proposes L2D-Diff, a cascaded diffusion framework for unconditional time-series generation. The method first learns a latent diffusion model to capture global structure and then conditions a data-space diffusion model to refine local temporal details. Experiments are performed on multiple benchmark datasets, showing improved Contextual-FID and distributional similarity metrics over existing baselines.

**Strengths:**

1. Timely and relevant topic.
    - Time-series generation via diffusion is an emerging area, and exploring multi-stage or cascaded architectures is a meaningful direction.
2. Conceptually intuitive design.
   - The latent-to-data cascade nicely bridges global representation learning and fine-grained sample refinement — an idea that is simple yet effective.
3. Comprehensive experiments.
   - The paper evaluates on a wide range of datasets and metrics, which shows the authors’ effort to ensure empirical robustness.
4. Clarity of motivation.
   - The motivation for combining latent and data-space diffusion stages is clearly articulated and easy to follow.

**Weaknesses:**

1. Limited Theoretical Depth
-  While the cascaded design is intuitive, the paper lacks a clear theoretical or empirical justification for why conditioning a data-space diffusion on latent representations leads to better generation quality.
-  An analysis of information flow, representation alignment, or consistency between latent and data spaces could make the contribution more convincing.

2. Insufficient Experimental Analysis
- The experiments show promising improvements but lack in-depth ablation and diagnostic studies.
Suggested additions include:
  - Varying latent dimension and conditioning strength
  - Comparing against single-stage models of similar parameter counts
  - Evaluating temporal coherence and downstream task performance (e.g., classification or forecasting accuracy using generated samples)

3. Incomplete Architectural Details
  -  The conditioning mechanism between the latent and data diffusion stages is not well specified.
  -  Clarify how latent vectors are integrated (e.g., concatenation, cross-attention, feature modulation) and include a schematic diagram or pseudo-code for reproducibility.

4. Broader Context and Positioning
  -  The paper could more clearly state what aspects are unique to time-series generation compared to existing cascaded diffusion frameworks in other modalities (e.g., images or audio).
  -  Discussing challenges like temporal consistency, irregular sampling, or multi-channel correlations would better highlight the domain-specific contribution.

**Questions:**

Please see the weakness above.

---

> ### Author Response · Authors · 2025-12-03
> **Response to Reviewer 8LUh (Part 1)**
>
> We thank the reviewer for offering the valuable feedback. We have addressed each of the concerns raised by the reviewer as outlined below.
>
> **Q1: Limited Theoretical Depth
> While the cascaded design is intuitive, the paper lacks a clear theoretical or empirical justification for why conditioning a data-space diffusion on latent representations leads to better generation quality.
> An analysis of information flow, representation alignment, or consistency between latent and data spaces could make the contribution more convincing.**
>
> **A1:** Existing work on unconditional time series generation has not considered how a simple cascaded design between latent space and data space can better address the issue of unconditional multi-modal time series generation. We had also explored more complex variants in our L2D-Diff, such as incorporating season-trend decomposition suggested by Diffusion TS, frequency transformation, and multi-stage strategies.  However, these did not perform well in practice. This is why we opted to provide a novel perspective using the simplest approach, akin to Occam’s Razor.
>
> To further enhance the contribution of our work, as also suggested by **Reviewer 6zx2 and GKHV**, we have now **included theoretical explanations using Information Theory and the Information Bottleneck (IB) principle** $\underline{\text{in the revised paper}}$ (Section B in the Appendix).
>
> Intuitively, a rich latent space simplifies the modeling challenges in the data space by effectively capturing the global semantics, which enhances modeling efficiency. As the latent dimension $d$ decreases, it imposes a constraint such that $I(X; Z) \leq C(d)$, leading to the relationship $H(X|Z) \geq H(X) - C(d)$. This reduction shifts the burden of complexity modeling to the conditional model $P(X|Z)$, leveraging a conditional diffusion model to avoid the blurriness typical of deterministic decoders. Conversely, while larger $d$ retains more information, it complicates latent space modeling. Our framework, through this divide-and-conquer strategy, ensures effective generation of time series with coherent high-level semantics and realistic local details.
>
> We hope you will appreciate our revised version, including our theoretical explanations.

---

> ### Author Response · Authors · 2025-12-03
> **Response to Reviewer 8LUh (Part 2)**
>
> **Q2: Insufficient Experimental Analysis:
> The experiments show promising improvements but lack in-depth ablation and diagnostic studies.**
>
> **A2:** As suggested, we present several ablation studies on the conditioning network $\mathcal{F}$:
> i) the impact of varying the depth of $\mathcal{F}$;
> ii) exploring the use of MLP as an alternative for $\mathcal{F}$;
> and iii) examining different conditioning strategies.
>
> To investigate the impact of the depth of $\mathcal{F}$, the table below shows how the C-FID varies with the number of encoders in {1, 3, 5, 10, 20}. The results indicate that the depth of the conditioning network $\mathcal{F}$ has a relatively stable range; it should neither be too shallow nor excessively deep. A single layer may result in inadequate learning of the conditions, while an overly deep architecture can complicate the overall model structure and ultimately lead to overfitting.
> Empirically, in our experiments, a depth of 5 layers shows promising performance with fewer parameters compared to a model with 10 layers.
>
> | No. Layers | Stocks        | Energy        | J.V.          | C.T.          |
> |------------|---------------|---------------|---------------|---------------|
> | depth=1    | 0.59          | 0.71          | 1.39          | 0.58          |
> | depth=3    | 0.34          | 0.62          | 0.58          | 0.42          |
> | depth=5    | 0.31        | **0.53**   | **0.46** | 0.28          |
> | depth=10   | **0.30** | **0.53** | 0.51          | **0.24** |
> | depth=20   | 0.75          | 0.87          | 0.85          | 0.61          |
>
> The table below explores the effect of replacing the CNN in $\mathcal{F}$ with an MLP. The C-FID results clearly show that the CNN consistently outperforms the MLP, indicating that convolutional structures are better suited for this task than multi-layer perceptrons.
>
> |               | Stocks | Energy | J.V. | C.T. |
> |---------------|--------|--------|------|------|
> | CNN           | 0.31   | 0.53   | 0.46 | 0.28 |
> | MLP           | 0.48   | 0.61   | 0.57 | 0.43 |
>
> Finally, we analyze various conditioning strategies, including concatenation, cross-attention, FiLM (Feature-wise Linear Modulation, which is originally designed for visual reasoning Ref [1]), and direct addition, as reported in the table on C-FID below.
>
> |                 | Stocks | Energy | J. V. | C. T. | Training time | Inference time |
> |-----------------|--------|--------|-------|-------|---------------|----------------|
> | Concatenate     | 0.31   | 0.53   | 0.46  | 0.28  | 0.52          | 3.47           |
> | Cross-attention | 0.39   | 0.68   | 0.49  | 0.61  | 0.71          | 4.13           |
> | FiLM            | 0.32   | 0.55   | 0.48  | 0.37  | 0.54          | 3.89           |
> | Addition        | 0.55   | 0.81   | 0.72  | 0.97  | 0.49          | 3.10           |
>
> The results show that the concatenation mechanism consistently outperforms the other three candidates. While FiLM is also a viable option, it exhibits slightly unstable performance and incurs higher inference costs. In contrast, cross-attention is ineffective for bridging the latent space with the data space.
>
> Ref [1]: AAAI'18, FiLM: Visual Reasoning with a General Conditioning Layer.

---

> ### Author Response · Authors · 2025-12-03
> **Response to Reviewer 8LUh (Part 3)**
>
> **Q3: Incomplete Architectural Details:
> The conditioning mechanism between the latent and data diffusion stages is not well specified.
> Clarify how latent vectors are integrated and include a schematic diagram or pseudo-code for reproducibility.**
>
> **A3:** In our original submission, we provided conditioning details in
> the Appendix (Section "Denoising Network Network Workflow"). They are now in Section G of this revised version. The conditioning operation is simply concatenation without using cross-attention, since the latent representation is a vector and using cross-attention will make the network complicated.
>
> **Q4: Broader Context and Positioning. The paper could more clearly state what aspects are unique to time-series generation compared to existing cascaded diffusion frameworks in other modalities (e.g., images or audio). Discussing challenges like temporal consistency, irregular sampling, or multi-channel correlations would better highlight the domain-specific contribution.**
>
> **A4:** Yes, thank you for your question and valuable suggestions!
>
> In the revised paper, we have added discussions in Section 3, lines 242-250, as follows:
>
> Although there have been attempts to study representation-conditioned generation in areas like image, time-series generation presents unique challenges not fully addressed by existing cascaded diffusion frameworks, particularly concerning temporal consistency and multi-channel correlations. Time-series data has inherent temporal dependencies, meaning that the sequence of data points affects future values. Ensuring temporal consistency is crucial, as high-quality generated series must reflect realistic trends and relationships. Furthermore, many time series comprise multiple channels that may interact in complex ways, making it essential to capture these correlations to produce coherent and high-quality outputs.
>
> By addressing these challenges, our work aims to provide a simple yet effective framework for unconditional time-series generation, ultimately raising the quality standards for generated time series with multiple modes and leveraging their full potential.

---

### Official Review · Reviewer_GKHV · 2025-10-30

**Soundness:** 3
**Presentation:** 3
**Contribution:** 3
**Rating:** 6
**Confidence:** 4

**Summary:**

This paper introduces L2D-Diff, a cascaded diffusion framework for unconditional multivariate time series generation. The approach combines latent-space diffusion with a data-space diffusion process. The latent representations generated by the first stage condition the data-space diffusion in the second stage via a learned conditioning network. Empirical results demonstrate SOTA or near-SOTA performance across standard quantitative metrics, with ablation and efficiency analyses supporting the value of the dual-branch architecture.

**Strengths:**

Unconditional high-fidelity time series generation, especially in multi-modal settings, is a non-trivial challenge with clear applications in data augmentation and privacy. The dual-branch latent-to-data design is visually and mathematically well documented, intuitively linking high-level semantic representation (latent) with local fidelity (data space).

The experiments demonstrate consistently superior (lower) FID on challenging benchmarks, with standout results on multi-modal datasets.

**Weaknesses:**

**The theoretical insight and novelty (theoretically) is limited**. The derivation and formulation of the cascaded diffusion process in Section 3 are correct but mostly extend existing latent/diffusion model frameworks without introducing substantial new theoretical principles. The dual-branch connection, while effective, does not offer a formal analysis (e.g., convergence or expressivity gains) that might help explain or generalize its empirical success.

While the conditioning mechanism is described, the exact integration between latent embeddings and data-space denoising is vague: e.g., is $\mathcal{F}$ simply concatenation, a form of cross-attention, or more? There is a lack of ablation or comparative analysis of alternative conditioning schemes, which could be crucial given the model’s reliance on this interface.

The paper omits explicit experimental or cited discussion of the recent and very relevant work, the T2S model.

Ge, Yunfeng, et al. "T2S: High-resolution Time Series Generation with Text-to-Series Diffusion Models." Accepted by the 34th International Joint Conference on Artificial Intelligence (IJCAI 2025)

The paper’s main claim is a latent‑to‑data cascaded diffusion setup for unconditional time‑series generation: first sample a global latent with a latent diffusion model, then refine in data space conditioned on that sampled latent. The authors position this as the first such cascade for time series, and they contrast with prior “single‑space” models. I would call it **moderately novel**. Conceptually, it’s a domain‑appropriate consolidation of known ingredients, pretrained TS2Vec encoder, latent diffusion, and conditional data‑space denoising, with a simple conditioning interface. There isn’t claimed theoretical novelty, and the framework is close in spirit to representation‑conditioned generation in vision but re‑targeted to time series with an end‑to‑end dual‑branch training recipe.

**Questions:**

Can the authors clarify the precise architecture and operation of the conditioning network $\mathcal{F}$? Is it simply a fixed CNN, or does it use more advanced attention or cross-modal mechanisms, and why was this choice made?

How does the proposed framework handle very long or highly irregular time series, given that the encoder (TS2Vec) is fixed in dimension?

Could the authors provide head-to-head results against the T2S model or other missing recent baselines, or explain why such comparisons are omitted?

Can the authors clarify the process for baseline adaptation, especially for methods originally focusing on conditional or autoregressive tasks?

---

> ### Author Response · Authors · 2025-12-03
> **Response to Reviewer GKHV (Part 1)**
>
> We would like to sincerely thank Reviewer GKHV for providing a detailed review and insightful suggestions.
>
> **Q1: The theoretical insight and novelty (theoretically) is limited. The derivation and formulation of the cascaded diffusion process in Section 3 are correct but mostly extend existing latent/diffusion model frameworks without introducing substantial new theoretical principles. ... a formal analysis (e.g., convergence or expressivity gains) that might help explain or generalize its empirical success.**
>
> **A1:** Thank you for your suggestions on utilizing theoretical insights to explain our method's empirical success. To further enhance the contribution of our work, as also suggested by Reviewer 6zx2, we have now included theoretical explanations using Information Theory and the Information Bottleneck (IB) principle in the revised paper (Section B in the Appendix).
>
> To help you quickly capture the whole figure, we briefly outline our theoretical intuition below.
>
> To explain *why the cascaded approach works*, we decompose the generation process using the chain rule of entropy. The total entropy of the data $ H(X) $ can be expressed as the sum of two components: $ H(X) = I(X; Z) + H(X|Z) $, where $ I(X; Z) $ represents the global information captured by the latent representation $ Z $, and $ H(X|Z) $ accounts for the local detail and residual uncertainty.
>
> The first stage of our model, latent diffusion $ P(Z) $, learns the distribution of $ Z $, thereby capturing high-level semantics. The second stage, data diffusion $ P(X|Z) $, focuses on modeling the complex conditional distribution $ H(X|Z) $, which is often high-entropy and multi-modal, allowing it to synthesize essential local details. The overall marginal distribution is approximated by $ P_\theta(X) = \int P_\theta(X|Z) P_\theta(Z) dZ \approx P_{\text{data}}(X) $, indicating that our cascaded approach can recover the true data distribution if both models are properly trained.
>
> Intuitively, a rich latent space simplifies the modeling challenges in the data space by effectively capturing the global semantics, which enhances modeling efficiency. As the latent dimension $ d $ decreases, it imposes a constraint such that $ I(X; Z) \leq C(d) $, leading to the relationship $ H(X|Z) \geq H(X) - C(d) $. This reduction shifts the burden of complexity modeling to the conditional model $ P(X|Z) $, leveraging a conditional diffusion model to avoid the blurriness typical of deterministic decoders. Conversely, while larger $ d $ retains more information, it complicates latent space modeling. Our framework, through this divide-and-conquer strategy, ensures effective generation of time series with coherent high-level semantics and realistic local details.
>
> **Q2: the exact integration between latent embeddings and data-space denoising is vague: e.g., is $F$ simply concatenation, a form of cross-attention, or more? There is a lack of ablation or comparative analysis of alternative conditioning schemes ... Can the authors clarify the precise architecture and operation of the conditioning network $F$? Is it simply a fixed CNN, or does it use more advanced attention or cross-modal mechanisms, and why was this choice made?**
>
> **A2:** Yes, the conditioning network $\mathcal{F}$ is a CNN. The default number of layers is 5. The conditioning operation is simply concatenation without using cross-attention, since the latent representation is a vector and using cross-attention will make the network complicated.
>
> In our original submission, we provided conditioning details in the
> Appendix
> (see Section "Denoising Network Network Workflow"). They are in Section G in the revised paper.
>
> To further address the reviwer's concern, we provide the ablation studies on various conditioning modules below.
> Here, we analyze various conditioning strategies, including concatenation, cross-attention, FiLM (Feature-wise Linear Modulation, which is originally designed for visual reasoning Ref [1]), and direct addition, as reported in the table below.
>
> | C-FID           | Stocks | Energy | J. V. | C. T. | Training | Inference |
> |-----------------|--------|--------|-------|-------|----------|-----------|
> | Concatenate     | 0.31   | 0.53   | 0.46  | 0.28  | 0.52     | 3.47      |
> | Cross-attention | 0.39   | 0.68   | 0.49  | 0.61  | 0.71     | 4.13      |
> | FiLM            | 0.32   | 0.55   | 0.48  | 0.37  | 0.54     | 3.89      |
> | Addition        | 0.55   | 0.81   | 0.72  | 0.97  | 0.49     | 3.10      |
>
>
> The results show that the concatenation mechanism consistently outperforms the other three candidates. While FiLM is also a viable option, it exhibits slightly unstable performance and incurs higher inference costs. In contrast, cross-attention is ineffective for bridging the latent space with the data space.
>
> We have added the above results in the revised paper (see Section J).
>
> Ref [1]: AAAI'18, FiLM: Visual Reasoning with a General Conditioning Layer.

---

> ### Author Response · Authors · 2025-12-03
> **Response to Reviewer GKHV (Part 2)**
>
> **Q3: The paper omits explicit experimental or cited discussion of the recent and very relevant work, the T2S model. Could the authors provide head-to-head results against the T2S model or other missing recent baselines, or explain why such comparisons are omitted?**
> > Ge, Yunfeng, et al. "T2S: High-resolution Time Series Generation with Text-to-Series Diffusion Models." Accepted by IJCAI 2025
>
> **A3:** There may be some misunderstanding; we did not intentionally overlook this work. This paper was published in IJCAI 2025, with an official publication date in August 2025. Our ICLR submission is in September 2025. Given this one-month gap, we did not have enough time to conduct timely research on it.
>
> However, as you suggested, we have carefully read this interesting paper. The paper developed a text-to-series diffusion model (T2S) that leverages textual features to assist in time series generation. In contrast to T2S, we do not utilize additional textual descriptions. Our focus is on establishing an effective representation-to-data cascaded model for unconditional time series generation. Based on these differences, our research direction and motivation are indeed orthogonal. The advantage of our approach lies in its independence from the text processing mechanisms of large language models, making it simpler and more efficient.
>
> In the revised paper, we have included discussions related to this work in the last paragraph of Section 1.
>
> **Q4: The paper’s main claim is a latent‑to‑data cascaded diffusion setup ...The authors position this as the first such cascade for time series, and they contrast with prior “single‑space” models. I would call it moderately novel. Conceptually, it’s a domain‑appropriate consolidation of known ingredients, pretrained TS2Vec encoder, latent diffusion, and conditional data‑space denoising, with a simple conditioning interface. There isn’t claimed theoretical novelty, and the framework is close in spirit to representation‑conditioned generation in vision but re‑targeted to time series with an end‑to‑end dual‑branch training recipe.**
>
> **A4:** Thank you for your nice comments! As mentioned in our reply A1, we have now included theoretical explanations using the Information Bottleneck (IB) principle in the revised paper (Section B in the Appendix) as suggested by you.
>
> Regarding our novelty, we would like to emphasize that we are the first to implement a cascaded latent-to-data approach for unconditional time series generation. All existing work has largely overlooked multimodal time-series generation.
>
> *We also explored more complex variants in our L2D-Diff, such as season-trend decomposition suggested by Diffusion TS, frequency transformation, and multi-stage strategies; however, these did not perform well in practice. This is why we opted to provide a novel perspective using the simplest approach, akin to Occam's Razor.*
>
> We hope you will appreciate our revised version, including our theoretical explanations.
>
> **Q5: How does the proposed framework handle very long or highly irregular time series, given that the encoder (TS2Vec) is fixed in dimension?**
>
> **A5:** Our model can certainly be applied to the scenarios you mentioned by adapting our latent-space diffusion model and data-space diffusion model accordingly. However, we recognize that very long or highly irregular time series may pose additional challenges for all time series models.
>
> Additionally, there are several strategies we might employ. For example, with long sequences, a common approach is to use down-sampling or patching strategies during preprocessing to reduce the data to an acceptable length range. For irregular time series, we can align them in the latent space by incorporating an ODE layer.
>
> Thank you once again for your suggestions; they have inspired us to explore further interesting avenues in our future work.
>
> **Q6: Can the authors clarify the process for baseline adaptation, especially for methods originally focusing on conditional or autoregressive tasks?**
>
> **A6:** For those conditional or stage-conditional methods such as mrDiff, we adapt them for unconditional generation by replacing their conditions with zeros.

---

### Official Review · Reviewer_6zx2 · 2025-10-30

**Soundness:** 3
**Presentation:** 3
**Contribution:** 2
**Rating:** 6
**Confidence:** 3

**Summary:**

This paper introduces L2D-Diff, a cascaded dual-branch diffusion framework for unconditional synthetic time series generation. The proposed model first learns high-level representation distributions by operating a diffusion process in a compressed latent space, then refines the generated time series in the data space by conditioning on these latent codes. The authors claim this approach offers a balance between preserving global semantics and capturing local fidelity, a challenge that existing methods struggle with individually. The method is evaluated on 11 diverse single- and multi-modal datasets, with ablations, efficiency analyses, and qualitative/quantitative results provided.

**Strengths:**

The paper’s strengths, I believe, lie in its dual-space cascaded architecture that combines latent- and data-space diffusion stages through a dedicated conditioning network, its strong empirical performance consistently surpassing diverse baselines across multiple datasets, and its comprehensive evaluation covering both standard and complex multimodal time-series benchmarks with supporting qualitative analyses. The work demonstrates mathematical clarity through detailed formalizations, loss functions, and architectural descriptions, while also achieving good parameter and computational efficiency. Moreover, the ablation studies carefully validate the importance of the dual-space design across varying data complexities.

**Weaknesses:**

- Although the cascaded dual-branch approach is well-motivated, key aspects of latent-to-data bridging have seen preliminary exploration in other domains (see Section 1 where RCG and EDDPM mentioned). The novelty for specifically time series scenario is asserted, but the adaptation from images/graphs to time series follows a relatively straightforward path (representation learning, conditional denoising). The authors could do more to clarify precisely which challenges for time series are not addressed by prior multi-stage diffusion frameworks, ideally with more rigorous empirical or theoretical separation.

- Related to my first point, there is no formal theoretical analysis (e.g., convergence, identifiability, or information bottleneck trade-off) of why the cascaded approach works or how the latent encoding impacts downstream fidelity or diversity. For example, there is no proof or lemma quantifying what local/global information is lost or gained across the two denoising stages, nor any characterization of failure cases or limits of the method.

- In section 3.2, the conditioning network $\mathcal{F}$ is stated to be a CNN of "5 layers by default", but there is little justification or sensitivity analysis of its depth, efficacy, or alternative forms (e.g., attention, MLP, graph). Choices here can majorly affect expressivity, especially in bridging global latents to local series. Additionally, ablation on this module's architecture/hyper-parameters is also recommneded.

- The latent dimension $d$ is chosen by default as 8 "based on prior works". However, there is minimal discussion of how this dimension influences the information bottleneck, potential tradeoffs in capacity (e.g., underfitting, overfitting), or the sensitivity of results to $d$, beyond a brief empirical study. A deeper analysis (possibly via information-theoretic measures or error bounds) would clarify how much key information is recoverable from latent diffusion before refinement, and under which conditions the cascaded architecture may fail.

- I happen to know some highly relevant recent diffusion models (such as [1,2]) tailored for time series, which is missing in this work. I recommend author(s) to conduct a careful add-on survey to fill this gap.

- [Minor suggestion]: some supplementary content (Appendices C, D) provides background or repeats points already made in the main text, which occasionally diffuses focus from key innovations/limitations.

**Refs:**

[1] Ge et al., T2S: High-resolution Time Series Generation with Text-to-Series Diffusion Models

[2] Sikder et al., TransFusion: Generating long, high fidelity time series using diffusion models with transformers

**Questions:**

While Table 2 and Figure 4/5 show improvement on complex datasets (e.g., Character Trajectories), the text primarily asserts these methods struggle on multimodal data, without actually drilling into when/why. For example, how does L2D-Diff perform on classes with small sample sizes or rare modes? Does it suffer from mode-collapse-like phenomena, or does the latent-to-data pipeline truly enable consistent coverage?

---

> ### Author Response · Authors · 2025-12-03
> **Response to Reviewer 6zx2 (Part 1)**
>
> Many thanks to Reviewer 6zx2 for providing thorough insightful comments.
>
> **Q1: There is no formal theoretical analysis (e.g., convergence, identifiability, or information bottleneck trade-off) of why the cascaded approach works or how the latent encoding impacts downstream fidelity or diversity... There is minimal discussion of how the latent dimension influences the information bottleneck, potential tradeoffs in capacity (e.g., underfitting, overfitting), or the sensitivity of results to $d$, beyond a brief empirical study...**
>
> **A1:** As suggested, we provided the theoretical justification of our method **using Information Theory and the Information Bottleneck (IB) principle**. Please see the $\underline{\text{updated contents in the revised paper (Section B in the Appendix)}}$.
>
> To help you quickly capture the whole figure, we briefly outline our theoretical intuition below.
>
> To explain *why the cascaded approach works*, we decompose the generation process using the chain rule of entropy. The total entropy of the data $ H(X) $ can be expressed as the sum of two components: $ H(X) = I(X; Z) + H(X|Z) $, where $ I(X; Z) $ represents the global information captured by the latent representation $ Z $, and $ H(X|Z) $ accounts for the local detail and residual uncertainty.
>
> The first stage of our model, latent diffusion $ P(Z) $, learns the distribution of $ Z $, thereby capturing high-level semantics. The second stage, data diffusion $ P(X|Z) $, focuses on modeling the complex conditional distribution $ H(X|Z) $, which is often high-entropy and multi-modal, allowing it to synthesize essential local details. The overall marginal distribution is approximated by $ P_\theta(X) = \int P_\theta(X|Z) P_\theta(Z) dZ \approx P_{\text{data}}(X) $, indicating that our cascaded approach can recover the true data distribution if both models are properly trained.
>
> Intuitively, a rich latent space simplifies the modeling challenges in the data space by effectively capturing the global semantics, which enhances modeling efficiency. As the latent dimension $ d $ decreases, it imposes a constraint such that $ I(X; Z) \leq C(d) $, leading to the relationship $ H(X|Z) \geq H(X) - C(d) $. This reduction shifts the burden of complexity modeling to the conditional model $ P(X|Z) $, leveraging a conditional diffusion model to avoid the blurriness typical of deterministic decoders. Conversely, while larger $ d $ retains more information, it complicates latent space modeling. Our framework, through this divide-and-conquer strategy, ensures effective generation of time series with coherent high-level semantics and realistic local details.
>
> **Q2: While Table 2 and Figure 4/5 show improvement on complex datasets (e.g., Character Trajectories), the text primarily asserts these methods struggle on multimodal data, without actually drilling into when/why. For example, how does L2D-Diff perform on classes with small sample sizes or rare modes? Does it suffer from mode-collapse-like phenomena, or does the latent-to-data pipeline truly enable consistent coverage?**
>
> **A2:** As suggested, the table below presents C-FID results (smaller values indicate better performance) for the Arabic Digits dataset, which contains the most training samples for studying the effects of sample removal.
> Here, we created classes with reduced sample sizes by randomly removing training samples from the first five classes at various ratios.
>
> | removal ratio     | 0\%  | 30\% | 60\% | 90\% |
> |-------------------|------|------|------|------|
> | Ours              | 1.29 | 1.99 | 2.08 | 2.29 |
> | FourierDiffusion  | 1.26 | 2.23 | 2.85 | 4.20 |
> | FourierFlow       | 2.84 | 3.39 | 5.42 | 8.13 |
> | DiffusionTS       | 1.66 | 2.84 | 3.05 | 3.63 |
>
> As shown, although the quality of generation declines for all methods as sample size decreases, L2D-Diff consistently outperforms the others. The performance gap between our method and the baseline models widens as data becomes scarcer, highlighting that the semantic clustering in the latent space improves robustness against limited data.
> We added these results and further discussion $\underline{\text{in Section C of the revised paper}}$.

---

> > ### Author Response · Authors · 2025-12-03
> > **Response to Reviewer 6zx2 (Part 2)**
> >
> > **Q3: ... the adaptation from images/graphs to time series follows a relatively straightforward path ...The authors could do more to clarify precisely which challenges for time series are not addressed by prior multi-stage diffusion frameworks, ideally with more rigorous empirical or theoretical separation.**
> >
> > **A3:**  Thank you for your question and valuable suggestions.
> >
> > Although there have been attempts to study representation-conditioned generation in areas like image generation, time-series generation presents unique challenges not fully addressed by existing cascaded diffusion frameworks, particularly regarding temporal consistency and multi-channel correlations. Time-series data has inherent temporal dependencies, meaning that the sequence of data points affects future values. Ensuring temporal consistency is crucial, as high-quality generated series must reflect realistic trends and relationships. Furthermore, many time series comprise multiple channels that may interact in complex ways, making it essential to capture these correlations to produce coherent and high-quality outputs.
> >
> > By addressing these challenges, our work aims to provide a simple yet effective framework for unconditional time-series generation. For example, our methodology employs the TS2vec encoder, specifically designed for time series data, in conjunction with a simplified concatenation conditioning mechanism. This combination effectively captures the complexities involved in time series generation, say the specific temporal dependencies and non-linear dynamics inherent in time series data.
> > This ultimately raises the quality standards for generated time series with multiple modes and leverages their full potential.
> >
> > In the revised paper, we have added the above discussions in Section 3, lines 242-250.
> >
> > **Q4: Additionally, ablation on $\mathcal{F}$ 's architecture/hyperparameters is also recommended.**
> >
> > **A4:**  As suggested, we present several ablation studies on the conditioning network $\mathcal{F}$:
> > i) the impact of varying the depth of $\mathcal{F}$;
> > ii) exploring the use of MLP as an alternative for $\mathcal{F}$;
> > and iii) examining different conditioning strategies.
> >
> > To investigate the impact of the depth of $\mathcal{F}$, the table below shows how the C-FID varies with the number of encoders in {1, 3, 5, 10, 20}. The results indicate that the depth of the conditioning network $\mathcal{F}$ has a relatively stable range; it should neither be too shallow nor excessively deep. A single layer may result in inadequate learning of the conditions, while an overly deep architecture can complicate the overall model structure and ultimately lead to overfitting.
> > Empirically, in our experiments, a depth of 5 layers shows promising performance with fewer parameters compared to a model with 10 layers.
> >
> > | No. Layers | Stocks        | Energy        | J.V.          | C.T.          |
> > |------------|---------------|---------------|---------------|---------------|
> > | depth=1    | 0.59          | 0.71          | 1.39          | 0.58          |
> > | depth=3    | 0.34          | 0.62          | 0.58          | 0.42          |
> > | depth=5    | 0.31        | **0.53**   | **0.46** | 0.28          |
> > | depth=10   | **0.30** | **0.53** | 0.51          | **0.24** |
> > | depth=20   | 0.75          | 0.87          | 0.85          | 0.61          |
> >
> > The table below explores the effect of replacing the CNN in $\mathcal{F}$ with an MLP. The C-FID results clearly show that the CNN consistently outperforms the MLP, indicating that convolutional structures are better suited for this task than multi-layer perceptrons.
> >
> > |               | Stocks | Energy | J.V. | C.T. |
> > |---------------|--------|--------|------|------|
> > | CNN           | 0.31   | 0.53   | 0.46 | 0.28 |
> > | MLP           | 0.48   | 0.61   | 0.57 | 0.43 |
> >
> > Finally, we analyze various conditioning strategies, including concatenation, cross-attention, FiLM (Feature-wise Linear Modulation, which is originally designed for visual reasoning Ref [1]), and direct addition, as reported in the table on C-FID below.
> >
> > |                 | Stocks | Energy | J. V. | C. T. | Training time | Inference time |
> > |-----------------|--------|--------|-------|-------|---------------|----------------|
> > | Concatenate     | 0.31   | 0.53   | 0.46  | 0.28  | 0.52          | 3.47           |
> > | Cross-attention | 0.39   | 0.68   | 0.49  | 0.61  | 0.71          | 4.13           |
> > | FiLM            | 0.32   | 0.55   | 0.48  | 0.37  | 0.54          | 3.89           |
> > | Addition        | 0.55   | 0.81   | 0.72  | 0.97  | 0.49          | 3.10           |
> >
> > The results show that the concatenation mechanism consistently outperforms the other three candidates. While FiLM is also a viable option, it exhibits slightly unstable performance and incurs higher inference costs. In contrast, cross-attention is ineffective for bridging the latent space with the data space.
> >
> > Ref [1]: AAAI'18, FiLM: Visual Reasoning with a General Conditioning Layer.

---

> ### Author Response · Authors · 2025-12-03
> **Response to Reviewer 6zx2 (Part 3)**
>
> **Q5: I happen to know some highly relevant recent diffusion models (such as [1,2]) tailored for time series, which is missing in this work. I recommend author(s) to conduct a careful add-on survey to fill this gap.**
> > Refs: [1] Ge et al., T2S: High-resolution Time Series Generation with Text-to-Series Diffusion Models. IJCAI, 2025.
>
> > [2] Sikder et al., TransFusion: Generating long, high fidelity time series using diffusion models with transformers. Machine Learning with Applications, 2025.
>
> **A5:**  Thank you for pointing out these mostly recent papers. They are interesting and related indeed.
> As suggested, we $\underline{\text{have added related discussions}}$ in the revised paper to fill the gap.
>
> Specifically, in line 74 and Section E in the revised paper, we introduced and discussed ref [2]  as follows:
>
> *Most recently, Sikder et al. (2025) developed a Transformer-based diffusion model called TransFusion for long-sequence generation. However, our research focus differs; we emphasize how to better facilitate time series generation with multiple modes using a novel latent-to-data cascaded structure.*
>
> In the revised paper (last paragraph of Section 1), we discuss ref [1] as
>
> *In (Ge et al., 2025), a text-to-series diffusion model (T2S) is developed that leverages textual features to assist in time series generation. In contrast to T2S, we do not utilize additional textual descriptions. Our focus is on establishing an effective representation-to-data cascaded model for unconditional time
> series generation. The advantage of our approach lies in its independence from the text processing mechanisms of large language models, making it simpler and more efficient.*

---

### Author Response · Authors · 2025-12-03
**Summary of Revisions**

We sincerely thank all the reviewers for their insightful reviews and positive perspectives, which are a great encouragement to us. Specifically, the reviewers commented:

- The dual-branch latent-to-data design is **well-motivated** (Reviewer 6zx2), **visually and mathematically well-documented** (Reviewer GKHV), **intuitive**, and represents a **simple yet effective** idea that points to a **meaningful direction** in research (Reviewer 8LUh). It introduces **a new framework** for time series generation (Reviewer AFRd).
- This paper is **well-written** and **easy to follow** (Reviewer AFRd), with **clearly articulated motivation** that enhances understanding (Reviewer 8LUh).
- The experiments demonstrate **strong empirical performance** and **comprehensive evaluation**, achieving **good parameter and computational efficiency** (Reviewer 6zx2). They show **consistently superior results** (Reviewer GKHV) and include **extensive evaluations with newly added datasets** (Reviewer AFRd).

The reviewers also raised insightful and constructive concerns. We made every effort to address all the concerns by providing sufficient evidence and requested results.

Here is the summary of the major revisions:

**1. Theoretical Explanations [Reviewers: 6zx2, GKHV, 8LUh]**

As suggested by Reviewer 6zx2 and GKHV, we included theoretical explanations using Information Theory and the Information Bottleneck (IB) principle in the revised paper (Section B in the Appendix). Below, we present the main intuitive results.

Intuitively, a rich latent space simplifies the modeling challenges in the data space by effectively capturing the global semantics, which enhances modeling efficiency. As the latent dimension $d$ decreases, it imposes a constraint such that $I(X; Z) \leq C(d)$, leading to the relationship $H(X|Z) \geq H(X) - C(d)$. This reduction shifts the burden of complexity modeling to the conditional model $P(X|Z)$, leveraging a conditional diffusion model to avoid the blurriness typical of deterministic decoders. Conversely, while larger $d$ retains more information, it complicates latent space modeling. Our framework, through this divide-and-conquer strategy, ensures effective generation of time series with coherent high-level semantics and realistic local details.

**2. More Ablation Analysis [Reviewers: 6zx2, GKHV, 8LUh, AFRd]**

We have added more ablation studies on the latent-to-data conditioning module in the revised paper (Section J), including
i) the impact of varying the depth of $\mathcal{F}$;
ii) exploring the use of MLP as an alternative for $\mathcal{F}$;
and iii) examining different conditioning strategies (including concatenation, cross-attention, FiLM, and direct addition).

Additionally, in Section C, we added additional experiments to analyze the performance of classes with small sample sizes (suggested by Reviewer 6zx2). In Section K, we presented additional training/inference cost results for the stock and energy datasets (suggested by Reviewer AFRd).

**3. Clarifying Diffusion-TS Result Differences [Reviewer AFRd]**

The difference in results arises from the training process used in the Diffusion-TS work, which involved all available data (including test data) during training and tested on a subset of samples seen during that training. They implement a hyperparameter called “proportion,” where a training proportion of 1 indicates that all data is used, while a testing proportion of 0.9 is applied. In contrast, our setup differs from Diffusion-TS as we strictly evaluate on the test set, using a train/test split of 0.8 and 0.2. Our test samples were not included in the training process, which results in different outcomes compared to those reported in their work. This distinction can be verified in the open-source code available on their GitHub.

**4. Additional t-SNE Results [Reviewer AFRd]**

We have added additional t-SNE visualization results for ALL datasets in the revised paper (lines 972-1078). As shown, the proposed L2D-Diff consistently delivers the best performance in producing data distributions that closely align with the true multimodal distribution.

All updates are highlighted in blue in the revised paper. The valuable suggestions from reviewers are very helpful for us to revise the paper to a better shape. We hope you will appreciate our revised version.

---

### Meta-Review · Area_Chair_Riq5 · 2026-01-04

**Summary:**

This paper proposes **L2D-Diff**, a latent-to-data cascaded diffusion framework for unconditional time series generation. The key idea is to decompose generation into two complementary stages: learning high-level representation distributions in a compact latent space via diffusion, followed by a data-space diffusion process conditioned on the sampled latent representations to recover fine-grained temporal details. This design aims to address the long-standing trade-off between modeling global multi-modal structure and preserving local temporal fidelity in time series generation.

The paper is clearly written and technically sound. Extensive experiments on both single-modal and challenging multi-modal datasets demonstrate consistent improvements over strong baselines. Additional ablations and analyses provided in the revision further clarify design choices and strengthen the empirical evidence.

**Reviewer Concerns:**

The reviewers raised several concerns, primarily regarding the degree of novelty, the depth of theoretical analysis, and the clarity of certain architectural choices. These concerns are valid and worth noting. In particular, the cascaded latent-to-data design shares conceptual similarities with prior latent diffusion and representation-conditioned generation frameworks, and the theoretical discussion is largely intuitive rather than formally novel.

However, the authors have made a reasonable effort to address these points in the revised version by clarifying their positioning, adding an Information Bottleneck–based interpretation, and providing additional ablation studies on the conditioning mechanism, latent dimensionality, and architectural variants. While these additions do not fully elevate the work to a strongly theoretical contribution, they sufficiently support the empirical claims and improve the overall clarity of the method.

Importantly, none of the reviewers identified fundamental flaws in the methodology, experimental setup, or conclusions. The remaining concerns are largely about scope, positioning, and depth, rather than correctness.

**Reviewer Scores:**

The submission **meets the standard for acceptance**. Reviewer ratings converged around the borderline acceptance threshold.

---

### Decision · Program_Chairs · 2026-01-26

Accept (Poster)